# Preserving Task-Relevant Information Under Linear Concept Removal

**Floris Holstege** [*†‡]   **Shauli Ravfogel**[*◇]   **Bram Wouters**[†]

University of Amsterdam, Department of Quantitative Economics[†]
New York University, Center for Data Science [◇]
Tinbergen Institute [‡]

## Abstract

Modern neural networks often encode unwanted concepts alongside task-relevant information, leading to fairness and interpretability concerns. Existing post-hoc approaches can remove undesired concepts but often degrade useful signals. We introduce SPLINCE—Simultaneous Projection for LINear concept removal and Covariance prEservation—which eliminates sensitive concepts from representations while exactly preserving their covariance with a target label. SPLINCE achieves this via an oblique projection that "splices out" the unwanted direction yet protects important label correlations. Theoretically, it is the unique solution that removes linear concept predictability and maintains target covariance with minimal embedding distortion. Empirically, SPLINCE outperforms baselines on benchmarks such as Bias in Bios and Winobias, removing protected attributes while minimally damaging main-task information.

## 1 Introduction

Deep neural networks (DNNs), including Language Models (LMs), have achieved great success in natural language processing (NLP) by learning rich representations of text, often referred to as embeddings [Cao, 2024, Wang et al., 2024a]. These embeddings were shown to also encode undesired information, such as markers of gender, leading to biased predictions [Bolukbasi et al., 2016]. In response, a variety of concept-removal methods has been developed to remove undesired information from embeddings. Examples of such methods are iterative nullspace projection (INLP, Ravfogel et al. [2020]), Linear adversarial concept erasure (RLACE, Ravfogel et al. [2022]), Spectral Attribute Removal (SAL, Shao et al. [2023]), and Least-squares Concept Erasure (LEACE, Belrose et al. [2023]). The shared objective of these methods is to make a concept—such as gender—-undetectable by any linear classifier, while preserving the original embeddings as much as possible.

Previous work has noted that a drawback of post-hoc concept-removal methods is that in addition to removing a particular concept, they tend to also eliminate other concepts and information from embeddings [Feder et al., 2021, Belinkov, 2022, Kumar et al., 2022, Guerner et al., 2025, Ravfogel et al., 2025]. Consider, for instance, a scenario where we wish to remove the effect of gender markers on a classifier that screens CVs for job applications. Naively applying concept-erasure techniques to removes gender markers from the input representations may inadvertently harm the model's performance on the primary task of profession prediction, since in real-world data, certain professions are strongly associated with gender. As a result, the erasure may distort relevant information, undermining both interpretability and utility.

In this paper, we seek to address a key drawback of post-hoc concept-removal methods. Our contribution is to introduce SPLINCE, a projection that (similar to LEACE or SAL) prevents any

---

[*]Equal contribution. Correspondence to f.g.holstege@uva.nl.

39th Conference on Neural Information Processing Systems (NeurIPS 2025).

linear classifier from predicting a concept, while also preserving the covariance with a task of interest. Mathematically, we construct an oblique projection that places the covariance between the representations and a protected attribute in its *kernel*, while maintaining the covariance between the representations and the main-task label in its *range*.

We prove that if a linear classifier is re-fitted after projection without regularization, *any* two projections that share the same kernel (i.e., that linearly erase the same subspace) will induce identical loss. In that sense, SPLINCE and previous methods such as LEACE [Belrose et al., 2023] (as well as more naive versions that do not explicitly aim to maintain the *minimality* of the projection) are all equivalent. For causal model interventions aiming to interpret its behavior—a situation where the underlying model is necessarily frozen—we argue that SPLINCE may perform a more surgical intervention, akin to minimizing the side effects of erasure on related concepts (e.g., removing *gender bias* while preserving *grammatical gender*). Empirically, we show that in a realistic classification setting, SPLINCE improves fairness in a highly challenging, imbalanced scenario, and removes stereotypes while maintaining correlated factual information.

## 2    Related work

**Concept-removal**: in response to growing concerns about DNNs relying on problematic or harmful concepts, a range of adversarial methods were developed to remove concepts from the embeddings of neural networks [Xie et al., 2017, Zhang et al., 2018]. However, these methods were later deemed unsuccessful at removing concepts [Elazar and Goldberg, 2018]. Subsequently, many works (including this) focused on preventing a linear classifier from predicting a concept as a more tractable alternative. This line of work is supported by the *linear subspace hypothesis* [Bolukbasi et al., 2016], which argues that concepts are represented in linear subspaces of embeddings (for a more elaborate discussion, see Park et al. [2024]).

**Existing linear concept-removal methods**: iterative nullspace projection (INLP, Ravfogel et al. [2020]) trains a linear classifier to predict a concept, and projects embeddings to the nullspace of the parameters of the linear classifier. This is repeated until the concept can no longer be predicted by the linear classifier. Relaxed Linear Adversarial Concept Erasure (RLACE, Ravfogel et al. [2022]) trains an orthogonal projection matrix such that a concept cannot be predicted by a linear classifier. These works were followed up by Least-squares Concept Erasure (LEACE, Belrose et al. [2023]), which ensures that no linear classifier can predict the concept (hereafter referred to as *linear guardedness*) while minimally altering the embeddings. Spectral attribute removal (SAL, Shao et al. [2023]) projects the embeddings orthogonal to the first $k$ eigenvectors of the covariance matrix $\mathrm{Cov}(\boldsymbol{x}, \boldsymbol{z})$. Mean Projection (MP, [Haghighatkhah et al., 2022]) projects embeddings to the nullspace of the mean difference between embeddings with and without concepts. It is equivalent to SAL when the concept is binary. SAL and MP also guarantee linear guardedness (similar to LEACE), whereas other methods may or may not satisfy this criterion.

**Linear concept-removal while preserving task-relevant information**: previous work suggests that linear concept-removal methods remove task-relevant information in addition to the concept they seek to remove [Belinkov, 2022, Kumar et al., 2022, Guerner et al., 2025]. In response, several alternatives have been proposed to address this issue, all removing a different linear subspace [Dev et al., 2021, Holstege et al., 2024, Bareeva et al., 2024, Shi et al., 2024]. However, each of these approaches sacrifices linear guardedness in order to retain more task relevant information. An alternative approach is to explicitly optimize for fairness while maintaining task performance [Shen et al., 2021]. However, this approach is more resource-intensive, as it cannot be applied to the frozen representations of a pretrained model. Moreover, because it modifies the original representations, it is unsuitable for scenarios where the intervention is intended to simulate causal experiments on the behavior of a pretrained LM, as discussed in section 4.2.

In this paper, we study how to retain task-relevant information while maintaining linear guardedness. Recent work has also focused on applying projections to parameters of DNNs instead of embeddings [Limisiewicz et al., 2024, 2025, Arditi et al., 2024]. This is outside of the scope of this paper.

# 3 Theory

We consider random vectors $\boldsymbol{x} \in \mathbb{R}^d$ and $\boldsymbol{z} \in \mathcal{Z}$. Here, $\boldsymbol{x}$ can be any vector of features, but should generally be thought of as embeddings of a deep neural network. In most cases we consider, they are the last-layer embeddings. The vector $\boldsymbol{z}$ represents the concept to be removed. It can be a binary or one-hot-encoded label, or continuous in the case of a regression setting.

The general idea of linear concept removal is to apply an affine transformation $r(\boldsymbol{x}) = \mathbf{P}\boldsymbol{x} + \boldsymbol{b}$, where $\mathbf{P} \in \mathbb{R}^{d \times d}$ and $\boldsymbol{b} \in \mathbb{R}^d$, that prevents classifiers from recovering the concept represented by $\boldsymbol{z}$ from the features $\boldsymbol{x}$. A special case of this objective aims to achieve *linear guardedness* [Ravfogel et al., 2023, Belrose et al., 2023], the inability of *linear* classifiers to predict the concept. Concretely, they show that linear guardedness is equivalent to zero covariance between the transformed features and the concept to be removed, i.e., $\mathrm{Cov}(r(\boldsymbol{x}), \boldsymbol{z}) = \mathbf{P}\boldsymbol{\Sigma}_{\boldsymbol{x},\boldsymbol{z}} = \mathbf{0}$, where $\boldsymbol{\Sigma}_{\boldsymbol{x},\boldsymbol{z}} = \mathrm{Cov}(\boldsymbol{x}, \boldsymbol{z})$ is the cross-covariance matrix of $\boldsymbol{x}$ and $\boldsymbol{z}$, and the symbol $\mathbf{0}$ can refer both to a zero vector and zero matrix. This condition, to which we will refer as the *kernel constraint*, only requires the kernel of $\mathbf{P}$ to contain the column space $\mathrm{colsp}(\boldsymbol{\Sigma}_{\boldsymbol{x},\boldsymbol{z}}) \subseteq \mathbb{R}^d$. The intuition is that $\mathbf{P}$ removes directions in the feature space that are linearly correlated with $\boldsymbol{z}$, making it impossible for linear classifiers to use the transformed features to predict $\boldsymbol{z}$. Importantly, this means that the requirement of linear guardedness does not uniquely determine the affine transformation. Belrose et al. [2023] use this freedom to minimize the impact of the transformation on the distance between the original and projected representations, driven by the intuition that the minimal-norm projection would minimally damage *other* semantic information encoded therein.

This problem turns out to have a closed-form solution: Belrose et al. [2023] show that for centered data, i.e., $\mathbb{E}[\boldsymbol{x}] = \mathbf{0}$, the constrained optimization problem

$$\underset{\mathbf{P} \in \mathbb{R}^{d \times d}}{\arg \min} \, \mathbb{E}\left[\left\|\mathbf{P}\boldsymbol{x} - \boldsymbol{x}\right\|_{\mathbf{M}}^2\right], \qquad \mathbf{P}\boldsymbol{\Sigma}_{\boldsymbol{x},\boldsymbol{z}} = \mathbf{0} \tag{1}$$

has solution $\mathbf{P}_{\text{LEACE}}^\star = \mathbf{W}^+ \mathbf{U}\mathbf{U}^\mathsf{T}\mathbf{W}$, where $\mathbf{W} = (\boldsymbol{\Sigma}_{\boldsymbol{x},\boldsymbol{x}}^{1/2})^+$ is a whitening matrix and $\mathbf{U}$ is a matrix whose orthonormal columns span the orthogonal complement of $\mathrm{colsp}(\mathbf{W}\boldsymbol{\Sigma}_{\boldsymbol{x},\boldsymbol{z}})$, which is the column space of the covariance matrix between $\boldsymbol{x}$ and $\boldsymbol{z}$ after whitening. Here, we denote by $\boldsymbol{\Sigma}_{\boldsymbol{x},\boldsymbol{x}} \in \mathbb{R}^{d \times d}$ the variance-covariance matrix of $\boldsymbol{x}$, by $\mathbf{A}^+$ the Moore-Penrose pseudoinverse of a matrix $\mathbf{A}$, and by $\mathbf{A}^{1/2}$ the p.s.d. square root of a p.s.d. matrix $\mathbf{A}$. The resulting transformation is an oblique projection with kernel $\mathrm{colsp}(\boldsymbol{\Sigma}_{\boldsymbol{x},\boldsymbol{z}})$ and range determined by the whitening matrix $\mathbf{W}$. The intuition is that this is the smallest possible kernel that satisfies the kernel constraint in equation 1, while the chosen range minimizes the distortion caused by a projection with this kernel.

## 3.1 SPLINCE: Ensuring linear guardedness while preserving task-relevant covariance

The LEACE projection ensures linear guardedness while minimizing the distortion of the features, but is oblivious of the main task of the model. A small expected norm squared may not optimally preserve information that is actually useful for the task at hand. Suppose now there is a random vector $\boldsymbol{y} \in \mathcal{Y}$ that represents the task-relevant information. Similar to the concept vector $\boldsymbol{z}$, it can be binary, one-hot encoded or continuous. We conjecture that task-relevant information in the features $\boldsymbol{x}$ is located in the directions that linearly covariate with $\boldsymbol{y}$. In other words, it is located in the column space of the covariance matrix $\boldsymbol{\Sigma}_{\boldsymbol{x},\boldsymbol{y}} = \mathrm{Cov}(\boldsymbol{x}, \boldsymbol{y})$. Indeed, *removing* this particular subspace completely prevents linear classification [Belrose et al., 2023].

In order to preserve this task-relevant information, we require the affine transformation $r(\boldsymbol{x}) = \mathbf{P}\boldsymbol{x} + \boldsymbol{b}$ not only to produce features that are linearly guarded for $\boldsymbol{z}$, but also to leave the covariance between $\boldsymbol{x}$ and $\boldsymbol{y}$ invariant. For this approach we cast the name SPLINCE (Simultaneous Projection for LINear concept removal and Covariance prEservation), which can be seen as an extension of LEACE. It eliminates sensitive concepts from representations while exactly preserving their covariance with a target label. The SPLINCE optimization problem is formulated in Theorem 1 and its solution is given by equation 4, which is the main theoretical contribution of this paper.

**Theorem 1.** *Let $\boldsymbol{x}$ and $\boldsymbol{z}, \boldsymbol{y}$ be random vectors with finite second moments, non-zero covariances between $\boldsymbol{x}$ and $\boldsymbol{z}$, and between $\boldsymbol{x}$ and $\boldsymbol{y}$, and $\mathbb{E}[\boldsymbol{x}] = \mathbf{0}$. Let $\mathbf{W} = (\boldsymbol{\Sigma}_{\boldsymbol{x},\boldsymbol{x}}^{1/2})^+$ be a whitening matrix. Define linear subspaces $\mathcal{U}^\perp = \mathrm{colsp}(\mathbf{W}\boldsymbol{\Sigma}_{\boldsymbol{x},\boldsymbol{z}})$ and $\mathcal{V} = \mathrm{colsp}(\mathbf{W}\boldsymbol{\Sigma}_{\boldsymbol{x},\boldsymbol{y}}) + \mathcal{U}^-$, where $\mathcal{U}^- = \mathcal{U} \cap (\mathrm{colsp}(\mathbf{W}\boldsymbol{\Sigma}_{\boldsymbol{x},\boldsymbol{z}}) + \mathrm{colsp}(\mathbf{W}\boldsymbol{\Sigma}_{\boldsymbol{x},\boldsymbol{y}}))^\perp$. Assume $\mathcal{U}^\perp \cap \mathrm{colsp}(\mathbf{W}\boldsymbol{\Sigma}_{\boldsymbol{x},\boldsymbol{y}}) = \{\mathbf{0}\}$. Then the*

*optimization problem*

$$\underset{\mathbf{P} \in \mathbb{R}^{d \times d}}{\arg\min} \mathbb{E}\left[\left\|\mathbf{P}\boldsymbol{x} - \boldsymbol{x}\right\|_{\mathbf{M}}^{2}\right] \tag{2}$$

*subject to the two constraints*

$$\mathbf{P}\boldsymbol{\Sigma}_{\boldsymbol{x},\boldsymbol{z}} = \mathbf{0}, \qquad and \qquad \mathbf{P}\boldsymbol{\Sigma}_{\boldsymbol{x},\boldsymbol{y}} = \boldsymbol{\Sigma}_{\boldsymbol{x},\boldsymbol{y}}, \tag{3}$$

*to be referred to as the kernel and range constraint, respectively, has the solution*

$$\mathbf{P}_{\mathrm{SPLINCE}}^{\star} = \mathbf{W}^{+}\mathbf{V}(\mathbf{U}^{\mathrm{T}}\mathbf{V})^{-1}\mathbf{U}^{\mathrm{T}}\mathbf{W}, \tag{4}$$

*where* $\mathbf{U}$ *and* $\mathbf{V}$ *are matrices whose orthonormal columns span* $\mathcal{U}$ *and* $\mathcal{V}$, *respectively.*

The proof of Theorem 1 is given in Appendix A.1. Compared to the LEACE optimization problem, the only difference is the additional condition of preserving task-relevant information, $\mathbf{P}\boldsymbol{\Sigma}_{\boldsymbol{x},\boldsymbol{y}} = \boldsymbol{\Sigma}_{\boldsymbol{x},\boldsymbol{y}}$, which we refer to as the *range constraint*. Similar to LEACE, $\mathbf{P}_{\mathrm{SPLINCE}}^{\star}$ is an oblique transformation with kernel $\mathrm{colsp}(\boldsymbol{\Sigma}_{\boldsymbol{x},\boldsymbol{z}})$. The difference lies in the range, which now contains $\mathrm{colsp}(\boldsymbol{\Sigma}_{\boldsymbol{x},\boldsymbol{y}})$ in order to fulfill the range constraint. Intuitively, the freedom that the LEACE optimization problem gives to the choice of the range is partially used to preserve the task-relevant information, i.e., the covariance between $\boldsymbol{x}$ and $\boldsymbol{y}$. The remainder of the freedom is used to minimize the distortion caused by the affine transformation, leading to whitening and unwhitening, similar to LEACE.

The main assumption of Theorem 1 is that $\mathcal{U}^{\perp} \cap \mathrm{colsp}(\mathbf{W}\boldsymbol{\Sigma}_{\boldsymbol{x},\boldsymbol{y}}) = \{\mathbf{0}\}$, which is satisfied as long as the subspaces spanned by $\mathrm{Cov}(\boldsymbol{x},\boldsymbol{z})$ and $\mathrm{Cov}(\boldsymbol{x},\boldsymbol{y})$ do not perfectly overlap. For the case where $\boldsymbol{z}$ and $\boldsymbol{y}$ are binary variables, this assumption is equivalent to requiring that the covariance vectors $\mathrm{Cov}(\boldsymbol{x},\boldsymbol{z})$ and $\mathrm{Cov}(\boldsymbol{x},\boldsymbol{y})$ are linearly independent (i.e., not proportional).

We note that, perhaps counter-intuitively, SPLINCE is not necessarily equivalent to LEACE if $\mathrm{colsp}(\boldsymbol{\Sigma}_{\boldsymbol{x},\boldsymbol{z}})$ and $\mathrm{colsp}(\boldsymbol{\Sigma}_{\boldsymbol{x},\boldsymbol{y}})$ are orthogonal subspaces. Only if those subspaces are orthogonal *after* whitening, SPLINCE and LEACE are equivalent. In that case the orthogonal projection of LEACE then already contains the task-relevant directions $\mathrm{colsp}(\mathbf{W}\boldsymbol{\Sigma}_{\boldsymbol{x},\boldsymbol{y}})$ in its range. We also note that, similar to LEACE [Belrose et al., 2023], in the case of non-centered data, i.e., $\mathbb{E}[\boldsymbol{x}] \neq \mathbf{0}$, the optimal affine transformation requires the addition of a constant $\boldsymbol{b}_{\mathrm{SPLINCE}}^{\star} = \mathbb{E}[\boldsymbol{x}] - \mathbf{P}_{\mathrm{SPLINCE}}^{\star}\mathbb{E}[\boldsymbol{x}]$. Finally, in Figure 1 we give a visual illustration of the steps of the projection matrix suggested by Theorem 1.

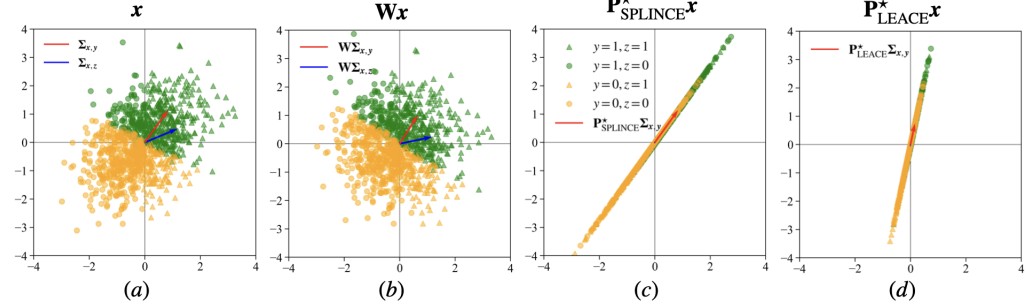

Figure 1: Illustration of the different steps for the projection suggested by Theorem 1 on two-dimensional data. The data $(a)$ is whitened $(b)$. Then, we use $\mathbf{V}(\mathbf{U}^{\mathrm{T}}\mathbf{V})^{-1}\mathbf{U}^{\mathrm{T}}$ to project parallel to $\mathbf{W}\boldsymbol{\Sigma}_{\boldsymbol{x},\boldsymbol{z}}$ onto $\mathbf{W}\boldsymbol{\Sigma}_{\boldsymbol{x},\boldsymbol{y}}$, and subsequently unwhiten $(c)$. With LEACE, the $\boldsymbol{\Sigma}_{\boldsymbol{x},\boldsymbol{y}}$ is altered $(d)$.

## 3.2 Last-layer linear concept removal with re-training

The first use case of SPLINCE we consider is linear concept removal applied to the embeddings of a DNN, after which a linear classifier is fitted on the transformed embeddings. This can be useful if it is demanded that a predictive model does not make use of sensitive concepts, like gender or race.

If we compare SPLINCE with other concept removal methods that guarantee linear guardedness, namely LEACE and SAL [Shao et al., 2023], we observe that they are all projections with the same kernel $\mathrm{colsp}(\boldsymbol{\Sigma}_{\boldsymbol{x},\boldsymbol{z}})$. They differ in the choice of the range. Interestingly, we find that the predictions of a linear classifier that is trained without regularization on the transformed embeddings are not affected by this choice of the range. In other words, all concept removal methods that ensure linear

guardedness will lead to the same predictions after re-training a linear classifier without regularization. We verify this empirically in Appendix B.1.

This result is formalized in Theorem 2, in which we consider training a model $f(\boldsymbol{x}; \boldsymbol{\theta})$ that only depends on the embeddings $\boldsymbol{x}$ and parameters $\boldsymbol{\theta}$ through their inner product. Examples of such models are linear and logistic regression, the latter typically being used as the linear classifier re-trained on the embeddings. Before fitting, a projection is applied to the embeddings. We consider two projections that have the same kernel, but different ranges. Theorem 2 shows that, in the case of a strictly convex loss function without regularization, both fitted models lead to the same predictions.

**Theorem 2** (Equivalent predictions after oblique re-training). *Consider observations $(\boldsymbol{x}, \boldsymbol{y}) \in \mathbb{R}^d \times \mathcal{Y}$ and a model $f(\boldsymbol{x}; \boldsymbol{\theta})$ that only depends on the inputs $\boldsymbol{x}$ and parameters $\boldsymbol{\theta}$ through their inner product, i.e., $f(\boldsymbol{x}; \boldsymbol{\theta}) = f(\boldsymbol{x}^{\mathrm{T}} \boldsymbol{\theta})$. Suppose we have data $\{(\boldsymbol{x}_k, \boldsymbol{y}_k)\}_{k=1}^{n}$, which is organized in a design matrix $\mathbf{X} \in \mathbb{R}^{n \times d}$ and $\mathbf{Y} \in \mathcal{Y}^n$. Before fitting the model, we apply an oblique transformation to the features $\boldsymbol{x}$. We consider two projections that have the same kernel $\mathcal{U} \in \mathbb{R}^d$, but different ranges $\mathcal{A}, \mathcal{B} \in \mathbb{R}^d$. We denote the corresponding transformation matrices as $\mathbf{P}_{\mathcal{A}}, \mathbf{P}_{\mathcal{B}}$, and we define $\boldsymbol{x}_{\mathcal{A}} = \mathbf{P}_{\mathcal{A}} \boldsymbol{x} \in \mathcal{A}$ and $\boldsymbol{x}_{\mathcal{B}} = \mathbf{P}_{\mathcal{B}} \boldsymbol{x} \in \mathcal{B}$. Let $\mathcal{L}(\mathbf{X}\boldsymbol{\theta}, \mathbf{Y})$ be a loss function with a unique minimizer. Then the following two minimizers*

$$\boldsymbol{\theta}_{\mathcal{A}}^* = \underset{\boldsymbol{\theta} \in \mathcal{A}}{\arg\min}\, \mathcal{L}(\mathbf{X}_{\mathcal{A}}\boldsymbol{\theta}, \mathbf{Y}), \quad \boldsymbol{\theta}_{\mathcal{B}}^* = \underset{\boldsymbol{\theta} \in \mathcal{B}}{\arg\min}\, \mathcal{L}(\mathbf{X}_{\mathcal{B}}\boldsymbol{\theta}, \mathbf{Y}) \tag{5}$$

*lead to the exact same predictions. In other words, $\boldsymbol{x}_{\mathcal{A}}^{\mathrm{T}} \boldsymbol{\theta}_{\mathcal{A}}^* = \boldsymbol{x}_{\mathcal{B}}^{\mathrm{T}} \boldsymbol{\theta}_{\mathcal{B}}^*$ for any $\boldsymbol{x} \in \mathbb{R}^d$.*

The proof of Theorem 2 is given in Appendix A.2. The intuition behind this result is that there exists an invertible linear transformation between the data after $\mathbf{P}_{\mathcal{A}}$ and the data after $\mathbf{P}_{\mathcal{B}}$. In other words, the choice of the range does not determine how much (linear) information about the target variable is lost in the oblique projection. This is solely determined by the choice of the kernel.

We point out that the assumption of unique minimizers corresponds to strictly convex loss functions, which is a common assumption for linear and logistic regression [Albert and Anderson, 1984]. In addition, note that the constraints in equation 5 of the parameters to $\mathcal{A}, \mathcal{B}$ is without loss of generality. Because of the inner product, components perpendicular to the subspaces do not affect predictions.

### 3.3 When does changing the range matter?

Theorem 2 provides a case where SPLINCE will lead to the same predictions as other concept removal methods that ensure linear guardedness (e.g., SAL and LEACE). Here, we identify two practical cases where applying projections with the same kernel and different ranges will typically lead to different predictions.

1. **When re-training the last layer with regularization**: if we include a regularization term (such as $||\boldsymbol{\theta}||_2$ or $||\boldsymbol{\theta}||_1$) in our loss function $\mathcal{L}$, then it no longer exclusively depends on the parameters via the inner product $\mathbf{X}\boldsymbol{\theta}$. This will generally lead to $\boldsymbol{x}_{\mathcal{A}}^{\mathrm{T}} \boldsymbol{\theta}_{\mathcal{A}}^* \neq \boldsymbol{x}_{\mathcal{B}}^{\mathrm{T}} \boldsymbol{\theta}_{\mathcal{B}}^*$ for any two projections with the same kernel and different ranges.

2. **When not re-training the last layer**: applying the same parameters to projected embeddings that lie in two different subspaces will typically not lead to the same predictions. If we consider projections $\mathbf{P}_{\mathcal{A}}$ and $\mathbf{P}_{\mathcal{B}}$ with ranges $\mathcal{A}$ and $\mathcal{B}$, then the predictions can only be the same if the parameters $\boldsymbol{\theta}^*$ lie in the orthogonal complement of both $\mathbf{P}_{\mathcal{A}}$ and $\mathbf{P}_{\mathcal{B}}$, i.e.,

$$\boldsymbol{x}^{\mathrm{T}} \mathbf{P}_{\mathcal{A}}^{\mathrm{T}} \boldsymbol{\theta}^* = \boldsymbol{x}^{\mathrm{T}} \mathbf{P}_{\mathcal{B}}^{\mathrm{T}} \boldsymbol{\theta}^* \quad \Leftrightarrow \quad \boldsymbol{x}^{\mathrm{T}} (\mathbf{P}_{\mathcal{A}} - \mathbf{P}_{\mathcal{B}})^{\mathrm{T}} \boldsymbol{\theta}^* = 0. \tag{6}$$

Since the projections considered in this paper are constructed without knowledge of the parameters, these will typically lie outside the orthogonal complements. Note that this use case is relevant for language modeling, where re-training the parameters of the last layer is typically not feasible in terms of computational resources and/or data availability.

For these cases we expect SPLINCE to outperform other methods that ensure linear guardedness, as it is designed to preserve task-relevant information. We empirically investigate this in the next section.[2]

---

[2]See this link for our code for the experiments, as well as an implementation of SPLINCE.

# 4 Experiments

This section is structured as follows. We start by investigating classification tasks when the last layer of the model is re-fitted with regularization. Then, in Section 4.2, we investigate language modeling, when the last layer of the language model (LM) is not re-trained post-projection. Finally, in order to qualitatively assess the effect of the different projections, we apply SPLINCE to black and white image data in Section 4.3. Across experiments, we compare SPLINCE to two other projections: LEACE and SAL (see Section 2). We focus on these projections since, similar to SPLINCE, they guarantee linear guardedness with regards to a concept, and only differ in choice of range. Furthermore, LEACE and SAL have been shown to outperform other existing concept-removal methods such as INLP and RLACE, which may or may not satisfy linear guardedness.

## 4.1 Classification where the last layer is re-trained with regularization

We focus on two classification problems. First, we use the *Bias in Bios* dataset on professions and biographies from De-Arteaga et al. [2019]. We focus on the set of biographies which carry the 'professor' label, $y_{\text{prof}} \in \{0, 1\}$, and seek to remove the concept of whether the subject was male or not, $z_{\text{gender}} \in \{0, 1\}$. Second, inspired by Huang et al. [2024], we use the *Multilingual Text Detoxification* dataset from Dementieva et al. [2024]. We focus on three languages (English, German and French), making the concept-label $z_{\text{lang}} \in \{1, 2, 3\}$ non-binary. This dataset consists of texts from users that are classified as toxic or non-toxic, $y_{\text{tox}} \in \{0, 1\}$.

**Set-up of the experiment**: we seek to investigate the impact of each projection as the correlation between the task of interest ($y_{\text{prof}}, y_{\text{tox}}$) and the concept to remove ($z_{\text{gender}}, z_{\text{lang}}$) becomes stronger. We expect that as the relationship becomes stronger, the difference between SPLINCE and other projections becomes greater. The reason is that stronger correlated labels have covariances with the embeddings that are typically more aligned. Removing the concept is then more likely to also remove information about the task of interest.

To alter the relationship between the task of interest and concept, we create smaller versions of the original datasets, where we vary the extent to which $y_{\text{prof}}, y_{\text{tox}}$ co-occur with respectively $z_{\text{gender}}, y_{\text{lang}}$. For the *Bias in Bios* dataset, we vary $p(y_{\text{prof}} = a \mid z_{\text{gender}} = a)$ with $a \in \{0, 1\}$, i.e., the conditional probability that the biography is of a professor and male or not a professor and female. For the *Multilingual Text Detoxification* dataset we vary $p(y_{\text{tox}} = 1 \mid z_{\text{lang}} = 1)$, e.g., the conditional probability that a toxic comment appears in the English language. We balance with respect to respectively $y_{\text{prof}}, y_{\text{tox}}$. In order to measure how much of the task-relevant information is retained after the projection, we create a test set where there is no correlation between $y_{\text{prof}}, y_{\text{tox}}$ and the respective concepts $z_{\text{gender}}, z_{\text{lang}}$. Additional details on the datasets are given in Appendix C.1.

**Models and training procedure**: for the *Bias in Bios* dataset, we finetune a BERT model [Devlin et al., 2019] to classify the profession. For the *Multilingual Text Detoxification* dataset we finetune multilingual E5 (ME5) embeddings Wang et al. [2024b] to classify the sentiment. For the BERT model, we add a linear layer on top of the embeddings of the [CLS] tokens for classification. For the ME5 embeddings, we add a linear layer on top of the average over all tokens. We apply projections to the last-layer embeddings - the [CLS] token for the BERT model, and the average over all tokens for the ME5 model. Afterwards we re-fit a logistic regression with $l_2$ regularization. We tune the strength of the $l_2$ regularization based on a validation set. This entire procedure (finetuning, projection, re-fitting, $l_2$ penalty selection) is repeated per seed. Additional details are given in Appendix C.2.

**Results**: the results of the experiments for both datasets are given in Figure 2. We focus on overall accuracy as well as worst-group accuracy. Worst-group accuracy is defined as the lowest accuracy for all combinations of the task and concept. A low worst-group accuracy reflects that a model relies on the correlation between the task and concept in the training data [Sagawa et al., 2020]. For both datasets, as the relationship between task and concept becomes stronger, SPLINCE outperforms the other projections in both accuracy and worst-group accuracy. As the correlation between the task and concept becomes stronger, SAL and LEACE remove a significant part of $\mathbf{\Sigma}_{x,y}$, contrary to SPLINCE. This is illustrated for the *Bias in Bios* dataset in Figure 7 in Appendix B.2

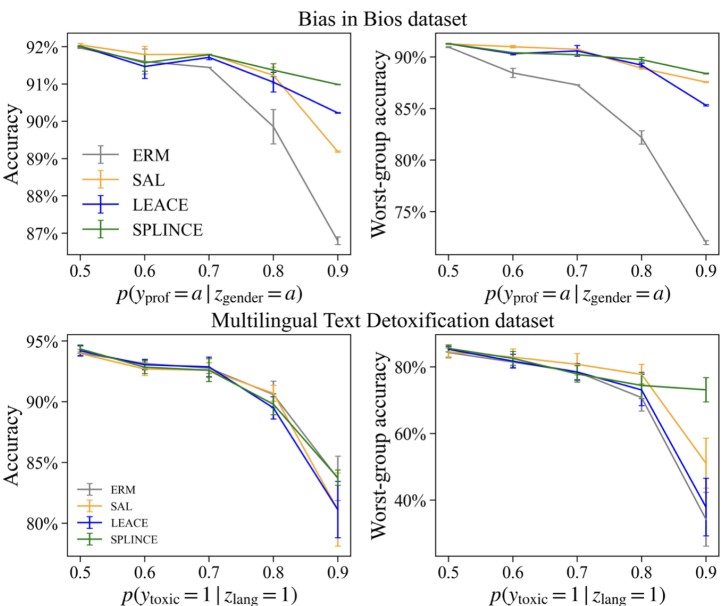

Figure 2: Performance of different projections on the *Bias in Bios* and *Multilingual Text Detoxification* dataset. We re-train the last-layer after applying each projection. Points are based on the average over 3 seeds, 5 seeds respectively for the two datasets. The error bars reflect the 95% confidence interval.

## 4.2 Language modeling

We focus on two language modelling tasks using the Llama series of models [Touvron et al., 2023, Grattafiori et al., 2024]. First, we use a dataset from Limisiewicz et al. [2024], which we refer to as the *profession dataset*. Inspired by Bolukbasi et al. [2016], this dataset contains prompt templates containing professions, which need to be finished by the LM (e.g.,'the plumber wanted that'). Each profession has a *stereotype* score $z_{\text{stereo}}$. This indicates how strongly a profession is connected with the male gender through stereotypical cues (e.g., plumber has a high stereotype score, while nurse has a low one). Each profession also has a *factual* score $y_{\text{fact}}$, which indicates how strong a profession is connected to the male gender through factual information (e.g., waiter has a high factual score, but waitress has a low one).

Second, we use the *Winobias* dataset from Zhao et al. [2018]. Each prompt contains two professions and pronouns. Prompts are marked pro-stereotypical or anti-stereotypical, denoted $z_{\text{pro-stereo}} \in \{0, 1\}$. In pro-stereotypical prompts, the coreference links to a profession with the stereotypical gender matching the gender of the pronoun. An example is 'The mechanic gave the clerk a present because he won the lottery. He refers to'. In anti-stereotypical cases, the profession's stereotypically assumed gender is different from the gender of the pronouns. The task is to finish the prompt with one of the 40 professions, with the correct profession denoted $y_{\text{profession}} \in \{1, 2, \dots 40\}$. For additional details on both datasets, see Appendix C.1.

**Set-up of the experiments**: for the *profession dataset*, our goal is to create an LM that does not rely on stereotypical cues, but on factual information. We estimate the extent to which the LM $\mathcal{M}$ relies on stereotypical cues or factual information as follows. Let $t_{\text{he}}/t_{\text{she}}$ be the tokens for "he"/"she". Let $\boldsymbol{t}_i$ denote tokens for a prompt $i$, and $p_{\mathcal{M}}(t_{\text{he}}|\boldsymbol{t}_i)$ the probability assigned by a model of the "he" token conditional on the prompt $\boldsymbol{t}_i$. We measure the log-odds ratio between the probability of the next token being 'he' or 'she' as

$$\text{odds}_{\text{he/she},i} = \log\left(\frac{p_{\mathcal{M}}(t_{\text{he}}|\boldsymbol{t}_i)}{p_{\mathcal{M}}(t_{\text{she}}|\boldsymbol{t}_i)}\right), \tag{7}$$

and estimate the linear regression

$$\text{odds}_{\text{he/she},i} = z_{\text{stereo},i}\hat{\beta}_{\text{stereo}} + y_{\text{fact},i}\hat{\beta}_{\text{fact}} + \hat{\alpha}. \tag{8}$$

Intuitively, the coefficients indicate to what extent the difference in the probability of assigning "he" or "she" can be explained by stereotypical cues or factual information [Limisiewicz et al., 2024].

For the *Winobias dataset*, we seek to create an LM that is able to provide the correct profession, regardless of whether or not the coreference link is pro-stereotypical. We seek to remove $z_{\mathrm{pro-stereo}}$ while preserving the covariance between the embeddings and $y_{\mathrm{profession}}$.

**Results**: for the experiment on the *profession dataset*, the results are shown in Table 1. We report the exponent of the coefficients in Equation 7, as this tells us how more likely the 'he' token becomes relative to the 'she' token after a one-unit increase in either the stereotypical or factual score. After applying any of the three projections, the extent to which the model relies on stereotypical information is greatly reduced, per the reduction in $\exp(\hat{\beta}_{\mathrm{stereo}})$. The extent to which the model relies on factual information after a projection is greatly reduced when applying SAL or LEACE, whereas it is increased or preserved after applying SPLINCE.

Table 1: Results of applying different projections to the last layer of various Llama models for the *profession dataset*.

| Model | Projection | $\exp(\hat{\beta}_{\mathrm{stereo}})$ | $\exp(\hat{\beta}_{\mathrm{fact}})$ |
|---|---|---|---|
| Llama 2 7B | Original | 3,59 | 15,71 |
|  | +SAL | 0,80 | 5,90 |
|  | +LEACE | 0,85 | 12,14 |
|  | +SPLINCE | 0,79 | 24,27* |
| Llama 2 13B | Original | 3,84 | 20,2 |
|  | +SAL | 0,84 | 4,81 |
|  | +LEACE | 0,88 | 16,32 |
|  | +SPLINCE | 0,81 | 33,24* |
| Llama 3 8B | Original | 3,98 | 19,02 |
|  | +SAL | 0,87 | 3,50 |
|  | +LEACE | 0,88 | 7,68 |
|  | +SPLINCE | 0,82 | 13,43* |

Note: the * indicates that difference between the factual coefficient of our projection and the factual coefficient of LEACE is statistically significant at the 1% level according to a one-tailed $t-$test. The exponent of the coefficients estimates how the odds ratio changes with a one-unit change in $z_{\mathrm{stereo}}$ and $y_{\mathrm{fact}}$, respectively.

For the experiment on the *Winobias dataset*, the results are shown in Figure 3. For two out of three Llama models, SPLINCE improves coreference accuracy more than the other projections. In particular, it strongly increases the accuracy for anti-stereotypical prompts. In Appendix B.6 report results for additional LM's outside of the Llama series.

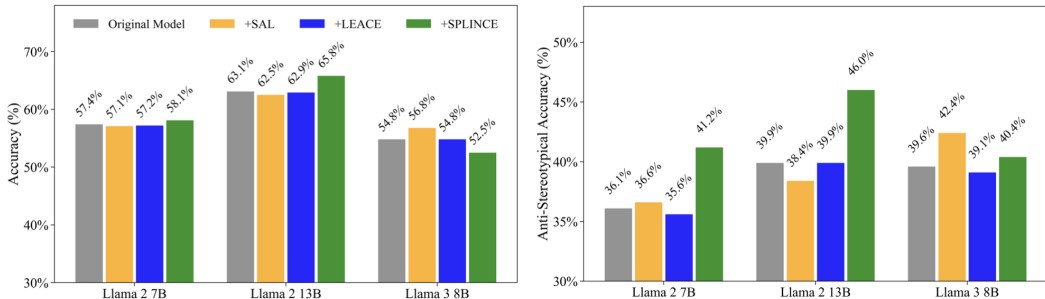

Figure 3: Results of applying different projections to the last layer of various Llama models for the *Winobias* dataset. The left plot shows the accuracy on a test set consisting of half pro-stereotypical and half anti-stereotypical prompts. The right plot shows the accuracy on the anti-stereotypical prompts in this test set.

## 4.3    Application to image data

We conduct an experiment for the *CelebA* dataset [Liu et al., 2015] that is similar to one from Ravfogel et al. [2022], Kleindessner et al. [2023], Holstege et al. [2024]. The goal of the experiment is to qualitatively show what features are removed by each projection. The concept to remove is whether or not someone is smiling, denoted $z_{\mathrm{smiling}} \in \{0, 1\}$, and we seek to preserve whether or not

someone wears glasses, $y_{\text{glasses}} \in \{0, 1\}$. We subsample 10,000 images from the original *CelebA* dataset such that $p(y_{\text{glasses}} = a \mid z_{\text{smiling}} = a) = 0.9$.

In Figure 4 we illustrate the effect of each projection on the raw pixels, for several images. SPLINCE accentuates parts of the image that are useful for distinguishing images with and without glasses. For instance, it tends to make the areas around the eyes lighter when someone does not wear glasses, and darken when they do. This shows that SPLINCE mitigates the damage to the "glasses" features exposed to a linear classifier, despite of the high correlation. We illustrate that this holds on average across the whole dataset in Appendix B.3.

In Appendix B.5 we include additional results CelebA dataset as introduced here, as well as the Waterbirds dataset Sagawa et al. [2020]. SPLINCE performs relatively worse for these vision classification tasks than the NLP classification tasks in 4.1. This gap in performance between the vision and NLP classification tasks is an interesting direction for future research.

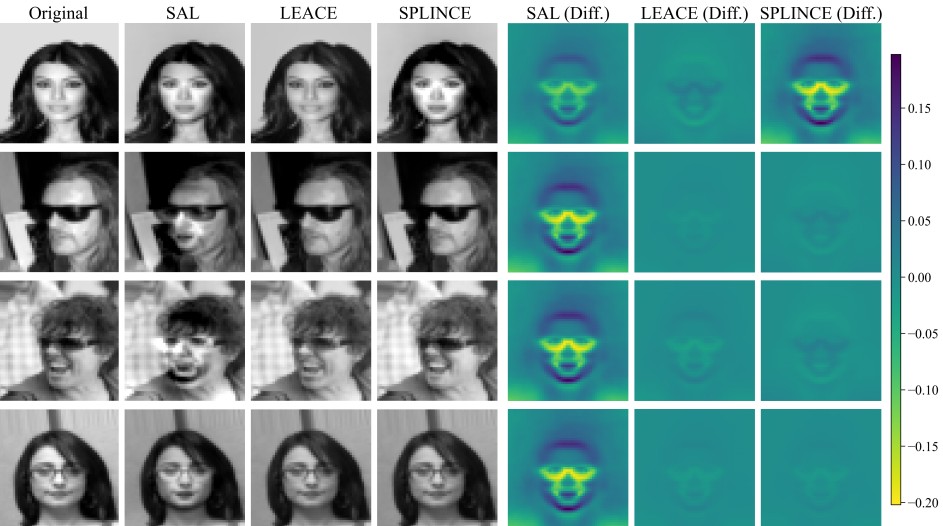

Figure 4: Application of different projections to raw pixel data of CelebA. The first columns shows the original image. The next four columns show the image, after the respective projection. The final three columns indicate the difference between the original image and the image after the projection.

## 5 Discussion and Limitations

Theoretically, we show that the range of the projection—particularly, whether or not it includes $\Sigma_{x,y}$—can matter only if one does not re-fit the last linear layer, or refit it with a regularization term. However, we lack a theoretical understanding that would explain under which conditions it *does* matter, and when it is expected to outperform LEACE. In Appendix A.3 we provide an initial investigation into this question, showing SPLINCE is guaranteed to outperform LEACE when we freeze the last layer and the embeddings are whitened. Future work should study whether preserving main-task covariance is optimal under more general settings. In this paper, we focus on preserving that quantity as it is intuitively related to main-task performance; however, we note that the SPLINCE objective can be modified to preserve *any* direction that is not identical to the covariance with the protected attribute.

In addition, a limitation of the SPLINCE objective is that it prioritizes preserving $\Sigma_{x,y}$ over minimizing $\mathbb{E}\left[\left\|\mathbf{P}x - x\right\|_{\mathbf{M}}^2\right]$. This can potentially lead to distortive changes to the embeddings. We investigate this limitation in Appendix B.4. As the SPLINCE objective sometimes fails when intervening in middle layers, we aim to study the adaption of the SPLINCE objective for preserving the directions in the representation space that are being used by an LM in some middle layer.

While we did not investigate a multi-modal setting (e.g. CLIP, Radford et al. [2021]), one potential limitation of SPLINCE is that covariance subspaces might not be aligned across modalities.

Finally, see Appendix D for a discussion on ethical considerations when applying SPLINCE.

## 6 Conclusion

We introduce SPLINCE, a method that generalizes previous concept-erasure methods by provably removing the ability to linearly predict sensitive information, while maintaining the covariance between the representations and *another* main-task label. Our analysis pins the problem down to a pair of geometric constraints—placing $\mathrm{colsp}(\Sigma_{x,z})$ in the kernel of the projection while forcing $\mathrm{colsp}(\Sigma_{x,y})$ to lie in its range—and proves that the oblique projector of Theorem 1 is the unique minimum-distortion solution under these constraints. Experimentally, SPLINCE tends to better preserve average and worst-group accuracy on the *Bias in Bios* and *Multilingual Text Detoxification* tasks when the task–concept correlation is high. In a language-modeling setting, we are able to influence stereotypical bias while preserving factual gender information; and in most models, it is better in preserving LM's ability to perform co-reference after debiasing in the *Winobias* dataset. Future work should formalize whether maintaining the covariance with the main-task translates into main-task loss guarantees and develop variants over the SPLINCE objective that impose weaker distortion when applied to earlier hidden layer, or performs better for vision datasets.

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

# A  Proofs of theorems

## A.1  Proof of theorem 1

We will prove Theorem 1 by making use of a basis tailored to the problem. In the end we will transform the resulting formula back to the basis-independent formulation of equation 4.

Let $m = \text{rank}(\boldsymbol{\Sigma}_{\boldsymbol{x},\boldsymbol{x}})$, with $1 \leq m \leq d$. Let $\mathcal{W} = \text{colsp}(\mathbf{W})$ be the subspace in $\mathbb{R}^d$ in which $\boldsymbol{x}$ has non-zero variance. Without loss of generality, we will define a basis for $\boldsymbol{x}$ where the first $m$ coordinates of $\boldsymbol{x}$ lie in $\mathcal{W}$. The last $l = d - m$ coordinates lie in its orthogonal complement $\mathcal{W}^\perp$. Our $\boldsymbol{x}$ can now be written as

$$\boldsymbol{x} = \begin{pmatrix} \tilde{\boldsymbol{x}} \\ \check{\boldsymbol{x}} \end{pmatrix}, \qquad \tilde{\boldsymbol{x}} \in \mathbb{R}^m, \check{\boldsymbol{x}} \in \mathbb{R}^l. \tag{9}$$

We will use $\tilde{x}_i \in \mathbb{R}$ to denote elements from the first $m$ coordinates, and $\check{x}_i \in \mathbb{R}$ to denote elements from the final $l$ coordinates.

We also assume that this basis is orthonormal with respect to the inner product $\mathbf{M}$, such that: $\boldsymbol{x}^\text{T}\mathbf{M}\boldsymbol{x} = \sum_{i=1}^d \alpha_i x_i^2$ for fixed $\alpha_1, ..., \alpha_d > 0$. Creating such a basis is always possible by standard orthogonalization procedures, now restricted to the coordinates of the respective subspaces $\mathcal{W}$ and $\mathcal{W}^\perp$. As a consequence of this, the optimization problem defined in Theorem 1 can be decomposed in $d$ independent optimization problems, one for each term in the sum that corresponds to the norm (squared), i.e., one for each coordinate of $\boldsymbol{x}$. To be more concrete, the optimization problem becomes for $i \in \{1, ..., d\}$

$$\underset{\mathbf{P} \in \mathbb{R}^{d \times d}}{\arg\min} \mathbb{E}\left[((\mathbf{P}\boldsymbol{x})_i - x_i)^2\right] \quad \text{subject to } \text{Cov}((\mathbf{P}\boldsymbol{x})_i, \boldsymbol{z}) = \mathbf{0},$$

$$\text{subject to } \text{Cov}((\mathbf{P}\boldsymbol{x})_i, \boldsymbol{y}) = \text{Cov}(x_i, \boldsymbol{y}), \tag{10}$$

where $x_i \in \mathbb{R}$ denotes the $i^{th}$ component of $\boldsymbol{x}$ and $(\mathbf{P}\boldsymbol{x})_i \in \mathbb{R}$ the $i^{th}$ component of $\mathbf{P}\boldsymbol{x}$. The weights $\alpha_1, ..., \alpha_d$ are left out, as they become irrelevant if we manage to find the minimum for each $x_i$.

**Lemma 1.** *Let*

$$\mathbf{P} = \begin{pmatrix} \tilde{\mathbf{P}} & \mathbf{0}_{m,l} \\ \mathbf{0}_{l,m} & \mathbf{0}_{l,l} \end{pmatrix}, \tag{11}$$

*where $\tilde{\mathbf{P}} \in \mathbb{R}^{m \times m}$ and where $\mathbf{0}_{m,l}$ is an $(m \times l)$-matrix of zeros and the other zero-matrices are defined similarly. A solution to the optimization problem, for $i \in \{1, ..., m\}$,*

$$\underset{\tilde{\mathbf{P}} \in \mathbb{R}^{d \times d}}{\arg\min} \mathbb{E}\left[((\tilde{\mathbf{P}}\tilde{\boldsymbol{x}})_i - \tilde{x}_i)^2\right] \quad \text{subject to } \text{Cov}((\tilde{\mathbf{P}}\tilde{\boldsymbol{x}})_i, \boldsymbol{z}) = \mathbf{0},$$

$$\text{subject to } \text{Cov}((\tilde{\mathbf{P}}\tilde{\boldsymbol{x}})_i, \boldsymbol{y}) = \text{Cov}(\tilde{x}_i, \boldsymbol{y}), \tag{12}$$

*corresponds then via equation 11 to a solution of the original optimization problem of Theorem 1.*

*Proof.* We start by dividing $\mathbf{P}$ in four block matrices,

$$\mathbf{P} = \begin{pmatrix} \tilde{\mathbf{P}} & \mathbf{B} \\ \mathbf{C} & \mathbf{D} \end{pmatrix}, \tag{13}$$

where $\tilde{\mathbf{P}} \in \mathbb{R}^{m \times m}, \mathbf{B} \in \mathbb{R}^{m \times l}, \mathbf{C} \in \mathbb{R}^{l \times m}, \mathbf{D} \in \mathbb{R}^{l \times l}$. We now proceed to determine these matrices, starting with a solution for $\mathbf{C}$ and $\mathbf{D}$. We do this by solving the optimization problem defined in equation 10 for the final $l$ rows. For $i \in \{l, ..., d\}$, we can write

$$(\mathbf{P}\boldsymbol{x})_i = \sum_{j=1}^m \text{C}_{p,j}\tilde{x}_j + \sum_{j=m+1}^d \text{D}_{p,j}\check{x}_j, \tag{14}$$

where $p = i - l + 1$. We use the indexation via $p$ because we use the rows $p \in \{1, ..., l\}$ of the matrices $\mathbf{C}$ and $\mathbf{D}$ respectively to represent $(\mathbf{P}\boldsymbol{x})_i$. The objective in equation 10 for $i \in \{l, ..., d\}$

and corresponding $p$ can be written as

$$\mathbb{E}\left[((\mathbf{P}\boldsymbol{x})_i - \check{x}_i)^2\right] = \mathbb{E}\left[\left(\sum_{j=1}^{m} \mathrm{C}_{p,j}\tilde{x}_j + \sum_{j=m+1}^{d} \mathrm{D}_{p,j}\check{x}_j - \check{x}_i\right)^2\right] \tag{15}$$

$$= \mathbb{E}\left[\sum_{j=1}^{m} \mathrm{C}_{p,j}\tilde{x}_j\right]^2 \tag{16}$$

$$= \left(\mathbf{C}\mathrm{Cov}(\tilde{\boldsymbol{x}}, \tilde{\boldsymbol{x}})\mathbf{C}^T\right)_{pp}, \tag{17}$$

where in the second equality we used that $\check{x}_i = 0$ almost surely and in the final equality we used that $\mathbb{E}[\tilde{x}_i] = 0$. We note that the values for $\mathrm{D}_{p,j}$ do not matter for the objective. Furthere, because $\mathrm{Cov}(\tilde{\boldsymbol{x}}, \tilde{\boldsymbol{x}})$ is p.s.d., we can achieve the minimum of equation 17 by setting $\mathbf{C} = \mathbf{0}$. For simplicity, we also set $\mathbf{D} = \mathbf{0}$. This then also trivially satisfies the kernel and range constraints for the components $i \in \{l, ..., d\}$.

For $i \in \{1, ..., m\}$, we can write

$$(\mathbf{P}\boldsymbol{x})_i = \sum_{j=1}^{m} \mathrm{A}_{i,j}\tilde{x}_j + \sum_{j=m+1}^{d} \mathrm{B}_{i,j}\check{x}_j. \tag{18}$$

We can set $\mathbf{B} = \mathbf{0}$ for the same reason as we chose $\mathbf{D} = \mathbf{0}$. The objective and constraints for $\tilde{\mathbf{P}}$ are then as in equation 12. This concludes the proof of Lemma 1. $\qquad\square$

In order to simplify the remaining objective for $\tilde{\mathbf{P}}$ in Lemma 1, we write $\tilde{\mathbf{P}} = \tilde{\mathbf{A}}\tilde{\mathbf{W}}$, where $\tilde{\mathbf{W}} = \boldsymbol{\Sigma}_{\tilde{\boldsymbol{x}},\tilde{\boldsymbol{x}}} \in \mathbb{R}^{m \times m}$ is full-rank, symmetric and p.s.d., and thus invertible. Because of this, optimizing for $\tilde{\mathbf{P}}$ is equivalent to optimizing for $\tilde{\mathbf{A}}$. Note that in this notation,

$$\boldsymbol{\Sigma}_{\boldsymbol{x},\boldsymbol{x}} = \begin{pmatrix} \boldsymbol{\Sigma}_{\tilde{\boldsymbol{x}},\tilde{\boldsymbol{x}}} & \mathbf{0}_{m,l} \\ \mathbf{0}_{l,m} & \mathbf{0}_{l,l} \end{pmatrix}, \tag{19}$$

an we can write the whitening matrix $\mathbf{W}$, and its Moore-Penrose inverse as

$$\mathbf{W} = \begin{pmatrix} \tilde{\mathbf{W}} & \mathbf{0}_{m,l} \\ \mathbf{0}_{l,m} & \mathbf{0}_{l,l} \end{pmatrix}, \quad \mathbf{W}^+ = \begin{pmatrix} \tilde{\mathbf{W}}^{-1} & \mathbf{0}_{m,l} \\ \mathbf{0}_{l,m} & \mathbf{0}_{l,l} \end{pmatrix}. \tag{20}$$

Using that we can write $\tilde{x}_i$ as

$$\tilde{x}_i = (\tilde{\mathbf{W}}^{-1}\tilde{\mathbf{W}}\tilde{\boldsymbol{x}})_i = \sum_{j=1}^{m} \tilde{\mathrm{W}}_{i,j}^{-1}(\tilde{\mathbf{W}}\tilde{\boldsymbol{x}})_j), \tag{21}$$

the remaining objective becomes, for $i \in \{1, ..., m\}$,

$$\mathbb{E}\left[((\mathbf{P}\boldsymbol{x})_i - \tilde{x}_i)^2\right] = \mathbb{E}\left[((\tilde{\mathbf{P}}\tilde{\boldsymbol{x}})_i - \tilde{x}_i)^2\right]$$

$$= \mathbb{E}\left[\left(\sum_{j=1}^{m}(\tilde{\mathrm{A}}_{i,j} - \tilde{\mathrm{W}}_{i,j}^{-1})(\tilde{\mathbf{W}}\tilde{\boldsymbol{x}})_j\right)^2\right]$$

$$= \mathrm{Var}\left(\sum_{j=1}^{m}(\tilde{\mathrm{A}}_{i,j} - \tilde{\mathrm{W}}_{i,j}^{-1})(\tilde{\mathbf{W}}\tilde{\boldsymbol{x}})_j\right) + \mathbb{E}\left[\sum_{j=1}^{m}(\tilde{\mathrm{A}}_{i,j} - \tilde{\mathrm{W}}_{i,j}^{-1})(\tilde{\mathbf{W}}\tilde{\boldsymbol{x}})_j\right]^2$$

$$= \mathrm{Var}\left(\sum_{j=1}^{m}(\tilde{\mathrm{A}}_{i,j} - \tilde{\mathrm{W}}_{i,j}^{-1})(\tilde{\mathbf{W}}\tilde{\boldsymbol{x}})_j\right)$$

$$= \sum_{h=1}^{m}\sum_{k=1}^{m}(\tilde{\mathrm{A}}_{i,h} - \tilde{\mathrm{W}}_{i,h}^{-1})(\tilde{\mathrm{A}}_{i,k} - \tilde{\mathrm{W}}_{i,k}^{-1})\mathrm{Cov}((\tilde{\mathbf{W}}\tilde{\boldsymbol{x}})_h, (\tilde{\mathbf{W}}\tilde{\boldsymbol{x}})_k)$$

$$= \left[\left(\tilde{\mathbf{A}} - \tilde{\mathbf{W}}^{-1}\right)\left(\tilde{\mathbf{A}} - \tilde{\mathbf{W}}^{-1}\right)^{\mathrm{T}}\right]_{i,i}, \tag{22}$$

where we used that $\mathbb{E}[(\tilde{\mathbf{W}}\tilde{\boldsymbol{x}})_j] = 0$ and $\mathrm{Cov}(\tilde{\mathbf{W}}\tilde{\boldsymbol{x}}, \tilde{\mathbf{W}}\tilde{\boldsymbol{x}}) = I_m$ by definition of the whitening matrix.

The constraints on $\tilde{\mathbf{P}}$ in the optimization problem in equation 12 translate into constraints on $\tilde{\mathbf{A}}$ and the optimization problem can be recast as

$$\tilde{\mathbf{A}}^* = \underset{\tilde{\mathbf{A}} \in \mathcal{C}_1 \cap \mathcal{C}_2}{\arg\min}\left\{\sum_{j=1}^{m}\left(\tilde{\mathbf{A}} - \tilde{\mathbf{W}}^{-1}\right)_{i,j}^2\right\}_{i=1}^{m}, \tag{23a}$$

where

$$\mathcal{C}_1 = \left\{M \in \mathbb{R}^{m \times m} \mid \tilde{\mathbf{W}}M\tilde{\mathbf{W}}\boldsymbol{\Sigma}_{\tilde{\boldsymbol{x}},\boldsymbol{z}} = \mathbf{0}_m\right\}, \tag{23b}$$

$$\mathcal{C}_2 = \left\{M \in \mathbb{R}^{m \times m} \mid \tilde{\mathbf{W}}M\tilde{\mathbf{W}}\boldsymbol{\Sigma}_{\tilde{\boldsymbol{x}},\boldsymbol{y}} = \tilde{\mathbf{W}}\boldsymbol{\Sigma}_{\tilde{\boldsymbol{x}},\boldsymbol{y}}\right\}. \tag{23c}$$

With more than one objective and two constraints, this is a constrained multiple optimization problem. We have seen before that it can be decomposed in $m$ separate constrained optimization problems. Each of these problems has a convex objective function and linear constraints, making the optimum $\tilde{\mathbf{A}}^*$ uniquely defined.

The constraints $\mathcal{C}_1$ and $\mathcal{C}_2$ can be interpreted as follows: $\tilde{\mathbf{A}}$ must be such that the columns of $\tilde{\mathbf{W}}\boldsymbol{\Sigma}_{\tilde{\boldsymbol{x}},\boldsymbol{z}}$ are in the kernel of $\tilde{\mathbf{W}}\tilde{\mathbf{A}}$ and that the columns of $\tilde{\mathbf{W}}\boldsymbol{\Sigma}_{\tilde{\boldsymbol{x}},\boldsymbol{y}}$ are eigenvectors of $\tilde{\mathbf{W}}\tilde{\mathbf{A}}$. This can be achieved by means of an oblique projection. If we define the following linear subspaces,

$$\tilde{\mathcal{U}}^{\perp} = \mathrm{colsp}\left(\tilde{\mathbf{W}}\boldsymbol{\Sigma}_{\tilde{\boldsymbol{x}},\boldsymbol{z}}\right), \tag{24}$$

$$\tilde{\mathcal{U}}^{-} = \tilde{\mathcal{U}} \cap \left(\mathrm{colsp}\left(\tilde{\mathbf{W}}\boldsymbol{\Sigma}_{\tilde{\boldsymbol{x}},\boldsymbol{z}}\right) + \mathrm{colsp}\left(\tilde{\mathbf{W}}\boldsymbol{\Sigma}_{\tilde{\boldsymbol{x}},\boldsymbol{y}}\right)\right)^{\perp}, \tag{25}$$

$$\tilde{\mathcal{V}} = \mathrm{colsp}\left(\tilde{\mathbf{W}}\boldsymbol{\Sigma}_{\tilde{\boldsymbol{x}},\boldsymbol{y}}\right) + \tilde{\mathcal{U}}^{-}, \tag{26}$$

and we define $\tilde{\mathbf{U}}$ and $\tilde{\mathbf{V}}$ as the matrices whose columns are an orthonormal basis of $\tilde{\mathcal{U}}$ and $\tilde{\mathcal{V}}$, respectively, then

$$\tilde{\mathbf{P}}_{\mathrm{obl}} = \tilde{\mathbf{V}}\left(\tilde{\mathbf{U}}^T\tilde{\mathbf{V}}\right)^{-1}\tilde{\mathbf{U}}^{\mathrm{T}} \tag{27}$$

is the transformation matrix of an oblique projection whose kernel is formed by the columns of $\tilde{\mathbf{W}}\boldsymbol{\Sigma}_{\tilde{\boldsymbol{x}},\boldsymbol{z}}$ and whose range include the columns of $\tilde{\mathbf{W}}\boldsymbol{\Sigma}_{\tilde{\boldsymbol{x}},\boldsymbol{y}}$. The latter means that the columns of $\tilde{\mathbf{W}}\boldsymbol{\Sigma}_{\tilde{\boldsymbol{x}},\boldsymbol{y}}$ are eigenvectors of $\tilde{\mathbf{P}}_{\mathrm{obl}}$.

We claim that any $\tilde{\mathbf{A}} \in \mathcal{C}_1 \cap \mathcal{C}_2$ can be written as $\tilde{\mathbf{B}}\tilde{\mathbf{P}}_{\mathrm{obl}}$, where $\tilde{\mathbf{B}}$ obeys the second constraint, i.e., $\tilde{\mathbf{B}} \in \mathcal{C}_2$. This identification is not unique, as multiple $\tilde{\mathbf{B}}$ lead to the same $\tilde{\mathbf{A}}$. We formalize this claim in the following lemma.

**Lemma 2.** *Let us define*

$$\mathcal{B}_{\tilde{\mathbf{P}}_{\text{obl}}} := \left\{ \tilde{\mathbf{B}}\tilde{\mathbf{P}}_{\text{obl}} \mid \tilde{\mathbf{B}} \in \mathcal{C}_2 \right\}. \tag{28}$$

*Then* $\mathcal{B}_{\tilde{\mathbf{P}}_{\text{obl}}} = \mathcal{C}_1 \cap \mathcal{C}_2$.

*Proof.* It is obvious that $\mathcal{B}_{\tilde{\mathbf{P}}_{\text{obl}}} \subseteq \mathcal{C}_1 \cap \mathcal{C}_2$, so we focus on proving that $\mathcal{C}_1 \cap \mathcal{C}_2 \subseteq \mathcal{B}_{\tilde{\mathbf{P}}_{\text{obl}}}$. For this, take an arbitrary $\mathbf{M} \in \mathcal{C}_1 \cap \mathcal{C}_2$. Let

$$\left\{ \text{colsp}(\tilde{\mathbf{W}}\Sigma_{\tilde{\boldsymbol{x}},\boldsymbol{z}}), \text{colsp}(\tilde{\mathbf{W}}\Sigma_{\tilde{\boldsymbol{x}},\boldsymbol{y}}), w_1, w_2, \ldots, w_k \right\}$$

be a basis of $\mathbb{R}^m$, where the $w_j \in \mathbb{R}^m$ are mutually orthonormal and orthogonal to $\tilde{\mathbf{W}}\Sigma_{\tilde{\boldsymbol{x}},\boldsymbol{z}}$ and $\tilde{\mathbf{W}}\Sigma_{\tilde{\boldsymbol{x}},\boldsymbol{y}}$. We then define a matrix $\tilde{\mathbf{B}} \in \mathbb{R}^{m\times m}$ in terms of its action on this basis, i.e.,

$$\begin{cases} \tilde{\mathbf{B}}\tilde{\mathbf{W}}\Sigma_{\tilde{\boldsymbol{x}},\boldsymbol{z}} = \mathbf{0}, \\ \tilde{\mathbf{B}}\tilde{\mathbf{W}}\Sigma_{\tilde{\boldsymbol{x}},\boldsymbol{y}} = \tilde{\mathbf{W}}^{-1}\tilde{\mathbf{W}}\Sigma_{\tilde{\boldsymbol{x}},\boldsymbol{y}}, \\ \tilde{\mathbf{B}}w_j = \mathbf{M}w_j, \qquad j = 1, 2, \ldots, k. \end{cases}$$

This implies that $\mathbf{M} = \tilde{\mathbf{B}}\tilde{\mathbf{P}}_{\text{obl}}$ and that $\tilde{\mathbf{B}} \in \mathcal{C}_2$. This concludes the proof of the lemma. □

Lemma 2 enables us to reformulate the optimization problem of equation 23 as follows. Define the set

$$\mathcal{B}^* = \arg\min_{\tilde{\mathbf{B}}\in\mathcal{C}_2} \left\{ \left[ \left( \tilde{\mathbf{B}}\tilde{\mathbf{P}}_{\text{obl}} - \tilde{\mathbf{W}}^{-1} \right) \left( \tilde{\mathbf{B}}\tilde{\mathbf{P}}_{\text{obl}} - \tilde{\mathbf{W}}^{-1} \right)^{\mathrm{T}} \right]_{i,i} \right\}_{i=1}^m. \tag{29}$$

This is a set of solutions to a different constrained optimization problem than equation 23. All solutions are equivalent in the sense that for any $\tilde{\mathbf{B}}_1^*, \tilde{\mathbf{B}}_2^* \in \mathcal{B}^*$ we have that $\tilde{\mathbf{B}}_1^*\tilde{\mathbf{P}}_{\text{obl}} = \tilde{\mathbf{B}}_2^*\tilde{\mathbf{P}}_{\text{obl}}$. Seen as an optimization for $\tilde{\mathbf{B}}\tilde{\mathbf{P}}_{\text{obl}}$, the objective is convex and the constraints are linear, making the optimum unique. Hence, $\tilde{\mathbf{A}}^* = \tilde{\mathbf{B}}^*\tilde{\mathbf{P}}_{\text{obl}}$ for any $\tilde{\mathbf{B}}^* \in \mathcal{B}^*$.

Now, we claim that $\tilde{\mathbf{W}}^{-1} \in \mathcal{B}^*$. The argument is that $\tilde{\mathbf{W}}^{-1}$ is a solution to the unconstrained equivalent of the optimization problem of equation 29 and, conveniently, also obeys the constraint $\mathcal{C}_2$. To see this, define

$$\mathcal{L}_i = \left[ \left( \tilde{\mathbf{B}}\tilde{\mathbf{P}}_{\text{obl}} - \tilde{\mathbf{W}}^{-1} \right) \left( \tilde{\mathbf{B}}\tilde{\mathbf{P}}_{\text{obl}} - \tilde{\mathbf{W}}^{-1} \right)^{\mathrm{T}} \right]_{i,i}, \tag{30}$$

for $i \in \{1, ..., m\}$, and take the derivative of the loss function with respect to elements of $\tilde{\mathbf{B}}$,

$$\frac{\partial \mathcal{L}_i}{\partial \tilde{\mathbf{B}}_{i,k}} = 2 \left( \left( \tilde{\mathbf{B}} - \tilde{\mathbf{W}}^{-1} \right) \tilde{\mathbf{P}}_{\text{obl}} \right)_{i,k}, \tag{31}$$

where we used that $\tilde{\mathbf{P}}_{\text{obl}}$ is idempotent. One solution to these first order conditions is $\tilde{\mathbf{B}} = \tilde{\mathbf{W}}^{-1}$. It is also obvious that $\tilde{\mathbf{W}}^{-1} \in \mathcal{C}_2$. Tracing the proof backwards, we conclude that

$$\mathbf{P}^* = \begin{pmatrix} \tilde{\mathbf{W}}^{-1}\tilde{\mathbf{P}}_{\text{obl}}\tilde{\mathbf{W}} & \mathbf{0}_{m,l} \\ \mathbf{0}_{l,m} & \mathbf{0}_{l,l} \end{pmatrix}, \tag{32}$$

solves the original constrained optimization problem of Theorem 1.

This expression is specific for the basis we chose at the beginning of the proof. If we let $\mathbf{S}$ be the matrix whose columns are the (orthonormal) vectors of the basis and we define

$$\mathbf{V} = \mathbf{S}\begin{pmatrix} \tilde{\mathbf{V}} \\ \mathbf{0} \end{pmatrix}, \quad \mathbf{U} = \mathbf{S}\begin{pmatrix} \tilde{\mathbf{U}} \\ \mathbf{0} \end{pmatrix}, \tag{33}$$

then we can write this result in a basis-independent way as

$$\mathbf{P}^* = \mathbf{W}^+ \begin{pmatrix} \tilde{\mathbf{P}}_{\text{obl}} & \mathbf{0}_{m,l} \\ \mathbf{0}_{l,m} & \mathbf{0}_{l,l} \end{pmatrix} \mathbf{W} = \mathbf{W}^+\mathbf{P}_{\text{obl}}\mathbf{W}, \qquad \text{with} \quad \mathbf{P}_{\text{obl}} = \mathbf{V}\left(\mathbf{U}^T\mathbf{V}\right)^{-1}\mathbf{U}^{\mathrm{T}}. \tag{34}$$

This corresponds to $\mathbf{P}^\star_{\text{SPLINCE}}$ in equation 4 and concludes the proof of Theorem 1.

## A.2 Proof of theorem 2

In order to prove theorem 2, we first prove the following Lemma.

**Lemma 3.** *Let* $\mathbf{P}_\mathcal{A}, \mathbf{P}_\mathcal{B} \in \mathbb{R}^{d \times d}$ *be (not necessarily orthogonal) projection matrices* $\mathbf{P}_\mathcal{A}^2 = \mathbf{P}_\mathcal{A}$ *and* $\mathbf{P}_\mathcal{B}^2 = \mathbf{P}_\mathcal{B}$ *with the* same *kernel* $\mathrm{Ker}(\mathbf{P}_\mathcal{A}) = \mathrm{Ker}(\mathbf{P}_\mathcal{B}) = \mathcal{U}$. *Set* $\mathcal{A} := \mathrm{Range}(\mathbf{P}_\mathcal{A})$, $\mathcal{B} := \mathrm{Range}(\mathbf{P}_\mathcal{B})$.

*Define*

$$F : \mathcal{A} \longrightarrow \mathcal{B}, \qquad F(\mathbf{P}_\mathcal{A}\boldsymbol{x}) := \mathbf{P}_\mathcal{B}\boldsymbol{x}, \quad \forall\, \boldsymbol{x} \in \mathbb{R}^d.$$

*Then* $F$ *is a* linear isomorphism*: it is well defined, linear, bijective, hence invertible.*

*Proof.* We prove the Lemma by showing $F$ is well defined, linear and bijective.
**Well defined.** If $\mathbf{P}_\mathcal{A}\boldsymbol{x} = \mathbf{P}_\mathcal{A}\boldsymbol{y}$, then $\mathbf{P}_\mathcal{A}(\boldsymbol{x} - \boldsymbol{y}) = \mathbf{0}$, so $\boldsymbol{x} - \boldsymbol{y} \in \mathcal{U} = \mathrm{Ker}(\mathbf{P}_\mathcal{B})$ and therefore $\mathbf{P}_\mathcal{B}(\boldsymbol{x} - \boldsymbol{y}) = \mathbf{0}$, i.e. $F(\mathbf{P}_\mathcal{A}\boldsymbol{x}) = F(\mathbf{P}_\mathcal{A}\boldsymbol{y})$.

**Linearity.** For $\boldsymbol{z}_1 = \mathbf{P}_\mathcal{A}\boldsymbol{x}_1$, $\boldsymbol{z}_2 = \mathbf{P}_\mathcal{A}\boldsymbol{x}_2$ and $\alpha \in \mathbb{R}$:

$$F(\boldsymbol{z}_1 + \boldsymbol{z}_2) = \mathbf{P}_\mathcal{B}(\boldsymbol{x}_1 + \boldsymbol{x}_2) = \mathbf{P}_\mathcal{B}\boldsymbol{x}_1 + \mathbf{P}_\mathcal{B}\boldsymbol{x}_2 = F(\boldsymbol{z}_1) + F(\boldsymbol{z}_2),$$
$$F(\alpha \boldsymbol{z}_1) = \mathbf{P}_\mathcal{B}(\alpha \boldsymbol{x}_1) = \alpha \mathbf{P}_\mathcal{B}\boldsymbol{x}_1 = \alpha F(\boldsymbol{z}_1).$$

**Injectivity.** If $F(\boldsymbol{z}_1) = F(\boldsymbol{z}_2)$, then $\mathbf{P}_\mathcal{B}\boldsymbol{x}_1 = \mathbf{P}_\mathcal{B}\boldsymbol{x}_2$ and $(\boldsymbol{x}_1 - \boldsymbol{x}_2) \in \mathcal{U} = \mathrm{Ker}(\mathbf{P}_\mathcal{A})$, so $\boldsymbol{z}_1 = \boldsymbol{z}_2$.

**Surjectivity.** Both $\mathcal{A}$ and $\mathcal{B}$ have dimension $d - \dim \mathcal{U}$; a linear, injective map between finite dimensional spaces of equal dimension is automatically surjective.

Thus $F$ is a linear isomorphism. $\qquad\qquad\square$

*Proof of Theorem 2.* Because $\mathrm{Ker}(\mathbf{P}_\mathcal{A}) = \mathrm{Ker}(\mathbf{P}_\mathcal{B}) = \mathcal{U}$, Lemma 3 provides an invertible linear map

$$F : \mathcal{A} \longrightarrow \mathcal{B}, \qquad F(\mathbf{P}_\mathcal{A}\boldsymbol{x}) = \mathbf{P}_\mathcal{B}\boldsymbol{x}.$$

**Step 1 — Transferring parameters.** For any $\boldsymbol{\theta}_\mathcal{A} \in \mathcal{A}$ define

$$\boldsymbol{\theta}_\mathcal{B} := F^{-\mathrm{T}}\boldsymbol{\theta}_\mathcal{A},$$

where $F^{-\mathrm{T}}$ denotes the transpose of the inverse map $F^{-1}$. Then, for every $\boldsymbol{x} \in \mathbb{R}^d$,

$$\underbrace{\boldsymbol{x}_\mathcal{B}^\mathrm{T}\boldsymbol{\theta}_\mathcal{B}}_{= (F\boldsymbol{x}_\mathcal{A})^\mathrm{T} F^{-\mathrm{T}}\boldsymbol{\theta}_\mathcal{A}} = \boldsymbol{x}_\mathcal{A}^\mathrm{T}\boldsymbol{\theta}_\mathcal{A},$$

so the two parameter vectors yield identical predictions.

**Step 2 — Empirical minimizers.** Let $\boldsymbol{\theta}_\mathcal{A}^* := \arg\min_{\boldsymbol{\theta} \in \mathcal{A}} \mathcal{L}(\mathbf{X}_\mathcal{A}\boldsymbol{\theta}, \mathbf{Y})$ be the (unique) minimizer over $\mathcal{A}$, and set $\boldsymbol{\theta}_\mathcal{B}^* := F^{-\mathrm{T}}\boldsymbol{\theta}_\mathcal{A}^*$. Because $\mathbf{X}_\mathcal{B} = F\mathbf{X}_\mathcal{A}$, the fitted predictions match:

$$\mathbf{X}_\mathcal{B}\boldsymbol{\theta}_\mathcal{B}^* = F\mathbf{X}_\mathcal{A}\boldsymbol{\theta}_\mathcal{A}^* = \mathbf{X}_\mathcal{A}\boldsymbol{\theta}_\mathcal{A}^*.$$

Hence both parameter choices achieve the same empirical loss value.

**Step 3 — Uniqueness over $\mathcal{B}$.** Since $\mathcal{L}$ has a *unique* minimizer on $\mathcal{B}$ and $\boldsymbol{\theta}_\mathcal{B}^*$ attains the minimal loss, it must coincide with the optimizer in equation 5. Therefore, for every input $\boldsymbol{x} \in \mathbb{R}^d$,

$$\boldsymbol{x}_\mathcal{A}^\mathrm{T}\boldsymbol{\theta}_\mathcal{A}^* = \boldsymbol{x}_\mathcal{B}^\mathrm{T}\boldsymbol{\theta}_\mathcal{B}^*,$$

which proves the theorem. $\qquad\qquad\square$

## A.3 Excess risk of SPLINCE vs. LEACE without re-training the last layer

In this section, we provide two theorems that show conditions under which SPLINCE is guaranteed to outperform LEACE when the last layer is frozen (e.g. not re-trained, contrasting the set-up of Theorem 2). In Theorem 3 we prove that for a linear regression with whitened data from any distribution, SPLINCE does not degrade the performance of a frozen last layer. In contrast, applying

LEACE may lead to an increase in the loss. In Theorem 4 we prove a similar statement, but for the case where the embeddings $\boldsymbol{x}$ follow a standard multivariate Gaussian distribution.

We emphasize that both theorems are limited in their applicability, as we generally would not expect last-layer embeddings to be whitened or follow a standard multivariate Gaussian distribution. Further work is required to understand more general conditions when SPLINCE might outperform LEACE.

**Theorem 3** (Excess-risk of LEACE and SPLINCE in a regression setting). *Let* $\mathbf{X} \in \mathbb{R}^{n \times d}$ *be whitened data with* $\boldsymbol{\Sigma}_{\boldsymbol{x},\boldsymbol{x}} = \boldsymbol{I}_d$ *and* $\mathbb{E}[\boldsymbol{x}] = \boldsymbol{0}_d$. *Assume the response model* $\mathbf{Y} = \mathbf{X}\boldsymbol{\beta} + \boldsymbol{\epsilon}$, *where* $\mathbb{E}[\boldsymbol{\epsilon}] = \boldsymbol{0}_n$, $\mathrm{Var}(\boldsymbol{\epsilon}) = \sigma^2 \boldsymbol{I}_n$ *and* $\mathbb{E}[\boldsymbol{\epsilon}^\top \mathbf{X}] = \boldsymbol{0}_d^{\mathrm{T}}$.

*Let* $\mathbf{Q}_{\mathrm{LEACE}}^\top := \boldsymbol{I}_d - \mathbf{P}_{\boldsymbol{z}}$, *where* $\mathbf{P}_{\boldsymbol{z}}$ *projects onto the subspace spanned by a concept variable* $\boldsymbol{z}$; *let* $\mathbf{Q}_{\mathrm{SPLINCE}}^\top := \mathbf{P}$ *be a projection satisfying* $\boldsymbol{\Sigma}_{\boldsymbol{x},\boldsymbol{y}} \in \mathrm{Range}(\mathbf{P})$. *Then:*

1. *The excess risk of* LEACE *is greater than or equal to zero,* $\Delta R(\mathbf{Q}_{\mathrm{LEACE}}) \geq 0$.

2. *The excess risk of* SPLINCE *is zero,* $\Delta R(\mathbf{Q}_{\mathrm{SPLINCE}}) = 0$.

*Proof.* For any square matrix $\mathbf{Q}$ define the *risk*

$$R(\mathbf{Q}) := \mathbb{E}\big[(\mathbf{Y} - \mathbf{X}\mathbf{Q}^\top \boldsymbol{\beta})^2\big],$$

and the *excess risk*

$$\Delta R(\mathbf{Q}) := R(\mathbf{Q}) - R(\boldsymbol{I}_d).$$

Using $\mathbf{Y} = \mathbf{X}\boldsymbol{\beta} + \boldsymbol{\epsilon}$ we have

$$\begin{aligned}
\mathbf{Y} - \mathbf{X}\mathbf{Q}^\top \boldsymbol{\beta} &= \mathbf{X}\boldsymbol{\beta} + \boldsymbol{\epsilon} - \mathbf{X}\mathbf{Q}^\top \boldsymbol{\beta} \\
&= \boldsymbol{\epsilon} + \mathbf{X}(\boldsymbol{I}_d - \mathbf{Q}^\top)\boldsymbol{\beta}.
\end{aligned} \tag{1}$$

Hence

$$\begin{aligned}
R(\mathbf{Q}) &= \mathbb{E}\Big[(\boldsymbol{\epsilon} + \mathbf{X}(\boldsymbol{I}_d - \mathbf{Q}^\top)\boldsymbol{\beta})^\top(\boldsymbol{\epsilon} + \mathbf{X}(\boldsymbol{I}_d - \mathbf{Q}^\top)\boldsymbol{\beta})\Big] \\
&= \underbrace{\mathbb{E}[\boldsymbol{\epsilon}^\top \boldsymbol{\epsilon}]}_{\sigma^2} + 2\underbrace{\mathbb{E}\big[\boldsymbol{\epsilon}^\top \mathbf{X}(\boldsymbol{I}_d - \mathbf{Q}^\top)\boldsymbol{\beta}\big]}_{0} + \mathbb{E}\Big[\boldsymbol{\beta}^\top(\boldsymbol{I}_d - \mathbf{Q})\mathbf{X}^\top \mathbf{X}(\boldsymbol{I}_d - \mathbf{Q}^\top)\boldsymbol{\beta}\Big] \\
&= \sigma^2 + \boldsymbol{\beta}^\top(\boldsymbol{I}_d - \mathbf{Q})\boldsymbol{\Sigma}_{\boldsymbol{x},\boldsymbol{x}}(\boldsymbol{I}_d - \mathbf{Q}^\top)\boldsymbol{\beta} \\
&= \sigma^2 + \|(\boldsymbol{I}_d - \mathbf{Q}^\top)\boldsymbol{\beta}\|_2^2, \tag{2}
\end{aligned}$$

because $\boldsymbol{\Sigma}_{\boldsymbol{x},\boldsymbol{x}} = \boldsymbol{I}_d$.

Subtracting $R(\boldsymbol{I}_d) = \sigma^2$ from (2) yields the expression

$$\boxed{\Delta R(\mathbf{Q}) = \|(\boldsymbol{I} - \mathbf{Q}^\top)\boldsymbol{\beta}\|_2^2}.$$

Take $\mathbf{Q} = \mathbf{Q}_{\mathrm{LEACE}} = \boldsymbol{I}_d - \mathbf{P}_{\boldsymbol{z}}$. Since $\mathbf{P}_{\boldsymbol{z}}$ is a projector, $(\boldsymbol{I}_d - \mathbf{P}_{\boldsymbol{z}})^\top = \boldsymbol{I}_d - \mathbf{P}_{\boldsymbol{z}}$; thus

$$\Delta R(\mathbf{Q}_{\mathrm{LEACE}}) = \|\mathbf{P}_{\boldsymbol{z}}\boldsymbol{\beta}\|_2^2.$$

Unless $\mathbf{P}_{\boldsymbol{z}}\boldsymbol{\beta} = \boldsymbol{0}_d$ (the degenerate case where $\boldsymbol{\beta}$ lies fully outside the concept subspace), this quantity is strictly positive.

Let $\mathbf{Q} = \mathbf{Q}_{\mathrm{SPLINCE}} = \mathbf{P}^\top$. By assumption $\boldsymbol{\Sigma}_{\boldsymbol{x},\boldsymbol{y}} \in \mathrm{Range}(\mathbf{P})$, and under whitening $\boldsymbol{\beta} = \boldsymbol{\Sigma}_{xy}$. Hence $\mathbf{P}\boldsymbol{\beta} = \boldsymbol{\beta}$ and $(\boldsymbol{I}_d - \mathbf{P}^\top)\boldsymbol{\beta} = (\boldsymbol{I}_d - \mathbf{P})\boldsymbol{\beta} = \boldsymbol{0}_d$. Applying the boxed identity, $\Delta R(\mathbf{Q}_{\mathrm{SPLINCE}}) = 0$. $\square$

**Theorem 4** (Excess-risk bound of LEACE and SPLINCE in a logistic-regression setting). *Let* $\mathbf{X} \in \mathbb{R}^{n \times d}$ *contain i.i.d. rows* $\boldsymbol{x} \sim \mathcal{N}(\boldsymbol{0}_d, \boldsymbol{\Sigma}_{\boldsymbol{x},\boldsymbol{x}})$. *Assume the conditional model*

$$\mathbb{P}(y = 1 \mid \boldsymbol{x}) = g(\mathbf{x}^\top \boldsymbol{\beta}), \qquad g(s) = \tfrac{1}{1+e^{-s}},$$

*with true parameter* $\boldsymbol{\beta} \in \mathbb{R}^d$.

*Define* $\mathbf{Q}_{\text{LEACE}}^{\top} := \boldsymbol{I}_d - \mathbf{P_z}$, *where* $\mathbf{P_z}$ *projects onto the subspace spanned by a concept variable* $\boldsymbol{z}$; *define* $\mathbf{Q}_{\text{SPLINCE}}^{\top} := \mathbf{P}$, *where* $\mathbf{P}$ *is an orthogonal projector satisfying* $\boldsymbol{\Sigma}_{\boldsymbol{x},\boldsymbol{y}} \in \text{Range}(\mathbf{P})$.

*Let* $\Delta_\ell(\mathbf{Q}) := \mathbb{E}[\ell(\mathbf{x}^{\top}\mathbf{Q}^{\top}\boldsymbol{\beta}, y) - \ell(\mathbf{x}^{\top}\boldsymbol{\beta}, y)]$ *denote the (population) excess logistic risk, with* $\ell(s, y) = \log(1 + e^s) - ys$.

*Then:*

1. *(LEACE)*
$$0 < \Delta_\ell(\mathbf{Q}_{\text{LEACE}}) \leq \frac{1}{8}\boldsymbol{\beta}^{\top}\mathbf{P_z}\,\boldsymbol{\Sigma}_{\boldsymbol{x},\boldsymbol{x}}\,\mathbf{P_z}\boldsymbol{\beta}.$$

   *In the whitened case* $\boldsymbol{\Sigma}_{\boldsymbol{x},\boldsymbol{x}} = \boldsymbol{I}_d$ *this reduces to* $\Delta_\ell(\mathbf{Q}_{\text{LEACE}}) \leq \frac{1}{8}\|\mathbf{P_z}\boldsymbol{\beta}\|_2^2$.

2. *(SPLINCE)* $\Delta_\ell(\mathbf{Q}_{\text{SPLINCE}}) = 0$.

*Proof.* Define

$$s := \boldsymbol{x}^{\top}\boldsymbol{\beta}, \qquad \hat{s} := \boldsymbol{x}^{\top}\mathbf{Q}^{\top}\boldsymbol{\beta}, \qquad \delta(x) := \hat{s} - s = \boldsymbol{x}^{\top}(\mathbf{Q}^{\top} - \boldsymbol{I}_d)\boldsymbol{\beta} = \boldsymbol{x}^{\top}(\boldsymbol{I}_d - \mathbf{Q}^{\top})(-\boldsymbol{\beta}).$$

**Three facts about the logistic loss.**

**Fact 1.** $\ell'(s, y) = \sigma(s) - y$.

**Fact 2.** $\ell''(s, y) = \sigma(s)(1 - \sigma(s)) \leq \frac{1}{4}$.

**Fact 3.** By the mean–value theorem, for some $\theta = \theta(x, y) \in (0, 1)$,

$$\ell(\hat{s}, y) = \ell(s, y) + \delta\,\ell'(s, y) + \frac{\delta^2}{2}\,\ell''(s - \theta\delta, y).$$

Taking expectations and using Fact 3,

$$\Delta_\ell(Q) = \mathbb{E}[\ell(\hat{s}, y) - \ell(s, y)] \tag{35}$$

$$= \mathbb{E}[\delta\,\ell'(s, y)] + \frac{1}{2}\,\mathbb{E}[\delta^2\ell''(s - \theta\delta, y)]. \tag{36}$$

Using $\mathbb{E}[y \mid x] = \sigma(s)$ (by model assumption),

$$\mathbb{E}[\delta\,\ell'(s, y)] = \mathbb{E}_x\Big[\delta(x)\,\underbrace{\mathbb{E}[\sigma(s) - y \mid x]}_{=0}\Big] = 0. \tag{37}$$

Fact 2 says $\ell'' \leq \frac{1}{4}$; hence

$$\frac{1}{2}\,\mathbb{E}[\delta^2\ell''(\cdot)] \leq \frac{1}{8}\,\mathbb{E}[\delta^2]. \tag{38}$$

Because $\boldsymbol{x} \sim \mathcal{N}(\boldsymbol{0}_d, \boldsymbol{\Sigma}_{\boldsymbol{x},\boldsymbol{x}})$,

$$\mathbb{E}[\delta^2] = \boldsymbol{\beta}^{\top}(\boldsymbol{I}_d - \mathbf{Q})\,\boldsymbol{\Sigma}_{\boldsymbol{x},\boldsymbol{x}}\,(\boldsymbol{I}_d - \mathbf{Q}^{\top})\boldsymbol{\beta}. \tag{39}$$

Thus

$$\boxed{0 \leq \Delta_\ell(\mathbf{Q}) \leq \frac{1}{8}\,\boldsymbol{\beta}^{\top}(\boldsymbol{I}_d - \mathbf{Q})\boldsymbol{\Sigma}_{\boldsymbol{x},\boldsymbol{x}}(\boldsymbol{I}_d - \mathbf{Q}^{\top})\boldsymbol{\beta}} \tag{40}$$

For Gaussian $x$ and differentiable $g$, Stein's lemma states $\mathbb{E}[g(x)\,x] = \boldsymbol{\Sigma}_{\boldsymbol{x},\boldsymbol{x}}\,\mathbb{E}[\nabla g(x)]$. Taking $g(x) = \sigma(\boldsymbol{x}^{\top}\boldsymbol{\beta})$ (a scalar function) gives

$$\boldsymbol{\Sigma}_{\boldsymbol{x},\boldsymbol{y}} = \mathbb{E}[\boldsymbol{x}y] = \mathbb{E}[\boldsymbol{x}\,\sigma(s)] = \boldsymbol{\Sigma}_{\boldsymbol{x},\boldsymbol{x}}\,\mathbb{E}[\sigma'(s)\,\boldsymbol{\beta}] \tag{41}$$

$$= \boldsymbol{\Sigma}_{\boldsymbol{x},\boldsymbol{x}}\,\boldsymbol{\beta}\,\underbrace{\mathbb{E}[\sigma(s)(1 - \sigma(s))]}_{=:C}. \tag{42}$$

Because $C > 0$, we can solve for the true parameter:

$$\boldsymbol{\beta} = \frac{1}{C}\,\boldsymbol{\Sigma}_{\boldsymbol{x},\boldsymbol{x}}^{-1}\boldsymbol{\Sigma}_{\boldsymbol{x},\boldsymbol{y}}. \tag{43}$$

Substituting yields for any $\mathbf{Q}$,

$$\Delta_\ell(\mathbf{Q}) \le \frac{1}{8C^2}\,\boldsymbol{\Sigma}_{\boldsymbol{x},\boldsymbol{y}}^{\top}\boldsymbol{\Sigma}_{\boldsymbol{x},\boldsymbol{x}}^{-1}(\boldsymbol{I}_d - \mathbf{Q})\boldsymbol{\Sigma}_{\boldsymbol{x},\boldsymbol{x}}(\boldsymbol{I}_d - \mathbf{Q}^{\top})\boldsymbol{\Sigma}_{\boldsymbol{x},\boldsymbol{x}}^{-1}\boldsymbol{\Sigma}_{\boldsymbol{x},\boldsymbol{y}}. \tag{44}$$

Take $\mathbf{Q} = \mathbf{Q}_{\text{LEACE}} = \boldsymbol{I}_d - \mathbf{P}_{\boldsymbol{z}}$. Since $(\boldsymbol{I}_d - \mathbf{Q}) = \mathbf{P}_{\boldsymbol{z}}$ is a projector,

$$\Delta_\ell(\mathbf{Q}_{\text{LEACE}}) \le \frac{1}{8C^2}\left\|\mathbf{P}_{\boldsymbol{z}}\boldsymbol{\Sigma}_{\boldsymbol{x},\boldsymbol{y}}\right\|^2.$$

Unless $\mathbf{P}_{\boldsymbol{z}}\boldsymbol{\Sigma}_{\boldsymbol{x},\boldsymbol{y}} = \mathbf{0}_d$ (degenerate), the RHS is strictly positive, proving the first half of the theorem.

Let $\mathbf{Q} = \mathbf{Q}_{\text{SPLINCE}} = \mathbf{P}^{\top}$ where $\mathbf{P}\boldsymbol{\Sigma}_{\boldsymbol{x},\boldsymbol{y}} = \boldsymbol{\Sigma}_{\boldsymbol{x},\boldsymbol{y}}$. Then $(\boldsymbol{I}_d - \mathbf{Q}^{\top})\boldsymbol{\Sigma}_{\boldsymbol{x},\boldsymbol{y}} = (\boldsymbol{I}_d - \mathbf{P})\boldsymbol{\Sigma}_{\boldsymbol{x},\boldsymbol{y}} = 0$, so the right-hand side of equation 44 is zero. Because $\Delta_\ell(\mathbf{Q}) \ge 0$ by definition, $\Delta_\ell(\mathbf{Q}_{\text{SPLINCE}}) = 0$. $\quad\square$

# B   Additional results

In this section, we report several results in addition to the experiments described in Section 4.

## B.1   Comparing the projections for different levels of regularization

In this subsection, we investigate how the difference in performance between SPLINCE, LEACE and SAL changes as a function of regularization. We use the *Bias in Bios* dataset and conduct the experiment as outlined in Section 4.1, but instead of selecting the $l_2$ regularization based on a validation set we fix the level or regularization. The results of this procedure are shown in Figure 5 and 6. As we lower the level of $l_2$ regularization (e.g. $\lambda = 0.0001$) the difference between the methods becomes indistinghuishable from zero.

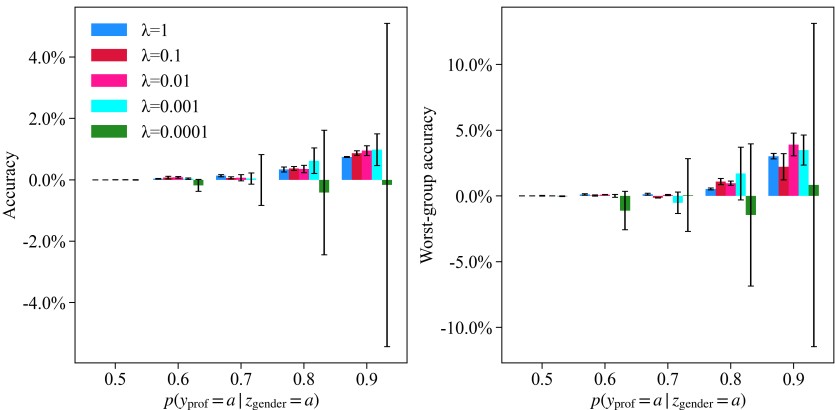

Figure 5: The difference between the SPLINCE and LEACE projections on the *Bias in Bios* dataset for different levels of $l_2$ regularization. We show the difference (worst-group) accuracy of SPLINCE minus the (worst-group) accuracy of LEACE. We re-train the last-layer after applying each projection. Points are based on the average over 3 seeds. The error bars reflect the 95% confidence interval.

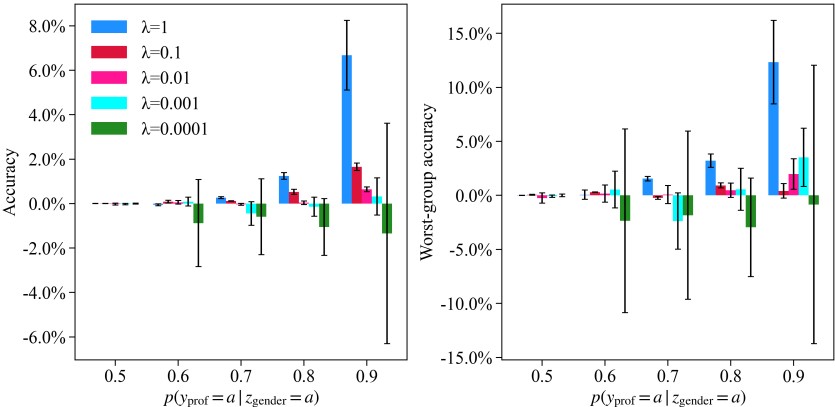

Figure 6: The difference between the SPLINCE and SAL projections on the *Bias in Bios* dataset for different levels of $l_2$ regularization. We show the difference (worst-group) accuracy of SPLINCE minus the (worst-group) accuracy of SAL. We re-train the last-layer after applying each projection. Points are based on the average over 3 seeds. The error bars reflect the 95% confidence interval.

## B.2 Removal of covariance for the *Bias in Bios* dataset for different projections

In this subsection, we briefly investigate the effect of different projections on the extent to which $\Sigma_{\boldsymbol{x}, y_{\text{prof}}}$ is preserved for the *Bias in Bios* dataset.

To quantify the extent to which $\Sigma_{\boldsymbol{x}, y_{\text{prof}}}$ is preserved, we measure the ratio of the squared $l_2$ norm of the transformed covariance $\mathbf{P}\Sigma_{\boldsymbol{x}, y_{\text{prof}}}$ after the projection $\mathbf{P}$ and original covariance $\Sigma_{\boldsymbol{x}, y_{\text{prof}}}$. Figure 7 shows the effect of changing the conditional probability $p(y_{\text{prof}} = a \mid z_{\text{gender}} = a)$ on this ratio. For SPLINCE, by design, $\Sigma_{\boldsymbol{x}, y_{\text{prof}}}$ is preserved regardless of $p(y_{\text{prof}} = a \mid z_{\text{gender}} = a)$, and the ratio remains 1. As $p(y_{\text{prof}} = a \mid z_{\text{gender}} = a)$ increases, SAL and LEACE lead to a removal of $\Sigma_{\boldsymbol{x}, y_{\text{prof}}}$. For instance, at $p(y_{\text{prof}} = a \mid z_{\text{gender}} = a) = 0.9$, after LEACE and SAL the ratio between the transformed and original covariances become respectively 0.07 and 0.001.

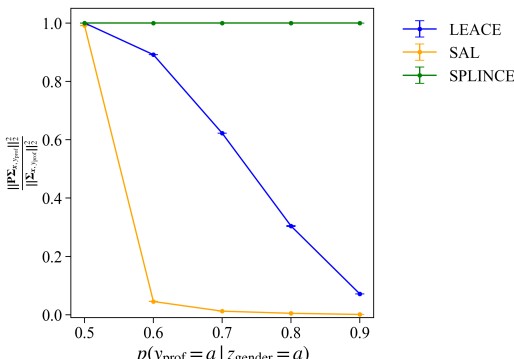

Figure 7: The ratio $\frac{||\mathbf{P}\Sigma_{\boldsymbol{x}, y_{\text{prof}}}||_2^2}{||\Sigma_{\boldsymbol{x}, y_{\text{prof}}}||_2^2}$ as a function of the relationship between $y_{\text{prof}}, z_{\text{gender}}$

## B.3 The mean difference between the original and transformed images for the *CelebA* dataset

To verify that the dynamics illustrated in Figure 4 hold across images, we measure the average difference between the original image before and after a projection. This is shown in Figure 8 for all combinations of $z_{\text{smiling}}$ and $y_{\text{glasses}}$. For individuals with glasses & not smiling, SPLINCE heavily accentuates the glasses by (on average) making them darker. For individuals without glasses & smiling, SPLINCE makes the area around the eyes lighter.

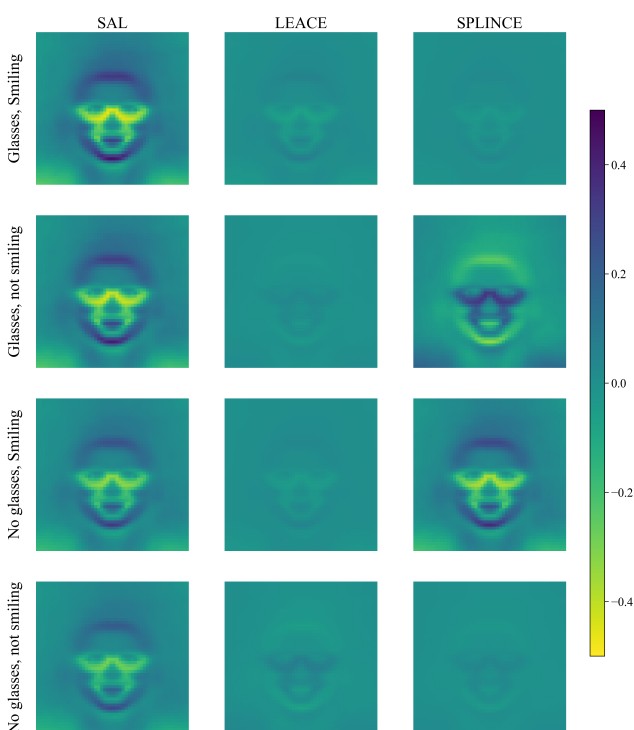

Figure 8: The mean difference between the original image and after a projection for every combination of $z_{\text{smiling}}, y_{\text{glasses}}$.

### B.4 Applying the projection to multiple layers

Previous work suggests to transform embeddings in multiple, earlier layers in order to amplify the effect of a projection (see, for instance Belrose et al. [2023], or Limisiewicz et al. [2024] for an example where parameters are adapted via projection). In this subsection, we repeat several of the experiments in Section 4 for different layers.

*Bias in Bios*: we apply the projections (SAL, LEACE, SPLINCE) to one of the 5 last layers of a BERT model. In this case, we do not re-train the subsequent layers. The accuracy after this procedure, per layer, is provided in Figure 9, and for worst-group accuracy in Figure 10. For later layers, similar to the results reported in Section 4.1, SPLINCE outperforms the other projections when the conditional probability $p(y_{\text{prof}} = a \mid z_{\text{gender}} = a)$ increases.

Per layer, we report the $||\mathbf{\Sigma}_{\boldsymbol{x}, z_{\text{gender}}}||_2$ and $||\mathbf{\Sigma}_{\boldsymbol{x}, y_{\text{prof}}}||_2$ in respectively Table 2 and 3. In the earlier layers (7-10), both $||\mathbf{\Sigma}_{\boldsymbol{x}, z_{\text{gender}}}||_2$ and $||\mathbf{\Sigma}_{\boldsymbol{x}, y_{\text{prof}}}||_2$ are relatively low, indicating relatively little covariance between the embeddings and $z_{\text{gender}}, y_{\text{prof}}$. As $||\mathbf{\Sigma}_{\boldsymbol{x}, z_{\text{gender}}}||_2$ and $||\mathbf{\Sigma}_{\boldsymbol{x}, y_{\text{prof}}}||_2$ increase in later layers (11-12), the difference between the projections also becomes clearer.

Table 2: The $||\mathbf{\Sigma}_{\boldsymbol{x}, z_{\text{gender}}}||_2$ for the biography dataset per layer

| Layer | $p(y_{\text{prof}} = a \mid z_{\text{gender}} = a)$ | | | | |
| | 0,5 | 0,6 | 0,7 | 0,8 | 0,9 |
|---|---|---|---|---|---|
| 7 | 0,30 | 0,33 | 0,33 | 0,33 | 0,32 |
| 8 | 0,42 | 0,44 | 0,44 | 0,43 | 0,43 |
| 9 | 0,33 | 0,34 | 0,35 | 0,35 | 0,36 |
| 10 | 0,34 | 0,37 | 0,36 | 0,33 | 0,34 |
| 11 | 1,07 | 1,14 | 1,10 | 1,09 | 1,02 |
| 12 | 1,37 | 1,45 | 1,47 | 1,52 | 1,81 |

Table 3: The $||\mathbf{\Sigma}_{\boldsymbol{x}, y_{\text{prof}}}||_2$ for the biography dataset per layer

| Layer | $p(y_{\text{prof}} = a \mid z_{\text{gender}} = a)$ | | | | |
| | 0,5 | 0,6 | 0,7 | 0,8 | 0,9 |
|---|---|---|---|---|---|
| 7 | 0,12 | 0,14 | 0,18 | 0,23 | 0,28 |
| 8 | 0,11 | 0,14 | 0,21 | 0,28 | 0,36 |
| 9 | 0,09 | 0,12 | 0,18 | 0,24 | 0,31 |
| 10 | 0,19 | 0,25 | 0,27 | 0,26 | 0,31 |
| 11 | 0,45 | 0,60 | 0,71 | 0,89 | 0,94 |
| 12 | 0,57 | 0,72 | 1,02 | 1,36 | 1,85 |

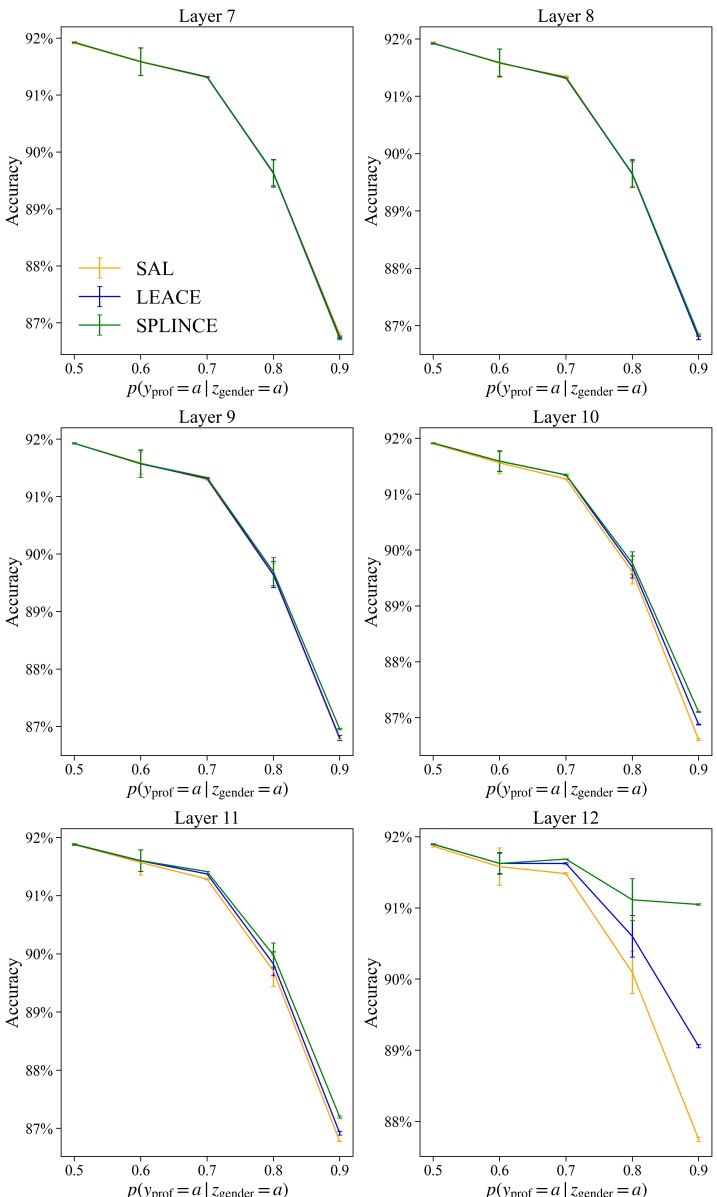

Figure 9: Average accuracy for different projections on the *Bias in Bios* dataset, for each of the 5 layers preceding the last layer. We do not re-train the subsequent layers after applying the projection. The points are the the average over 3 seeds. The error bars reflect with the 95% confidence interval.

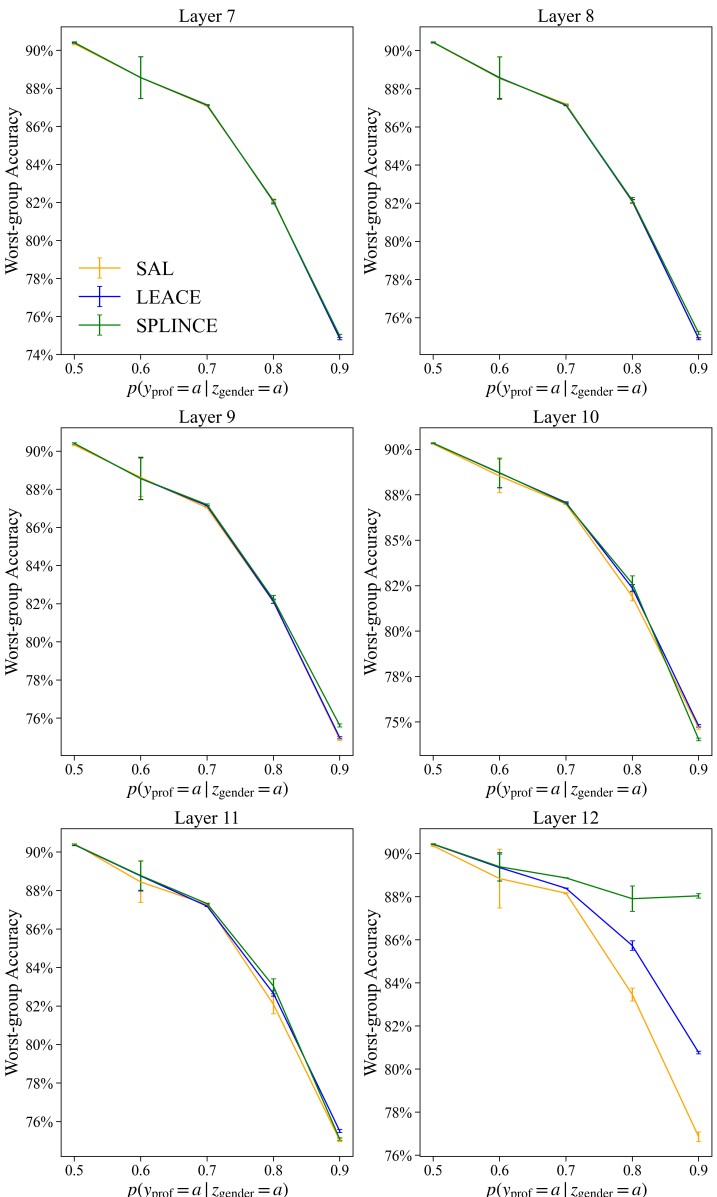

Figure 10: Worst-group accuracy for different projections on the Biography dataset, for the 5 layers preceding the last layer. We do not re-train the last-layer after applying the projection. The points are the the average over 3 seeds. The error bars reflect with the 95% confidence interval.

*Profession dataset*: When applying the projections, we start by the first layer in the sequence of layers where we alter the embeddings. After determining the projection for the embeddings at this layer, we subsequently determine it for the next, taking into account the projection of the previous layer. Table 4 shows the result of this procedure on Llama 2 7B. It remains the case that after SPLINCE applying SPLINCE to multiple layers, the model relies to a greater extent on factual information than when using SAL or LEACE.

Table 4: Results of applying different projections to different layers for the *profession dataset* on Llama 2 7B.

| Model | Layers | Method | $\exp(\hat{\beta}_{\text{stereo}})$ | $\exp(\hat{\beta}_{\text{fact}})$ |
|---|---|---|---|---|
| | | Original | 3,59 | 15,71 |
| | Last 3 | SAL | 1,14 | 4,07 |
| | | LEACE | 1,04 | 14,30 |
| | | SPLINCE | 1,00 | 37,94* |
| | | Original | 3,59 | 15,71 |
| Llama 2 7B | Last 5 | SAL | 0,86 | 5,29 |
| | | LEACE | 0,63 | 14,35 |
| | | SPLINCE | 0,64 | 15,09* |
| | | Original | 3,59 | 15,71 |
| | Last 9 | SAL | 1,18 | 6,48 |
| | | LEACE | 0,90 | 26,12 |
| | | SPLINCE | 0,87 | 78,17* |

Note: the * indicates that the difference between the factual coefficient of our projection and the factual coefficient of LEACE is statistically significant at the 1% level according to a one-tailed $t-$test. The exponent of the coefficients estimates how the odds ratio changes with a one-unit change in $z_{\text{stereo}}$ and $y_{\text{fact}}$, respectively.

*Winobias dataset*: Similar to the *Profession* dataset, we apply the projections to embeddings of subsequent layers, taking into account the projection at the previous layer. As illustrated per Figure 11, we observe that the performance of SPLINCE decreases as we apply it to more layers. Potentially, this is because of the (large) dimensionality of $\Sigma_{\boldsymbol{x}, y_{\text{profession}}} \in \mathbb{R}^{d \times 40}$. This result highlights a potential limitation of our projection.

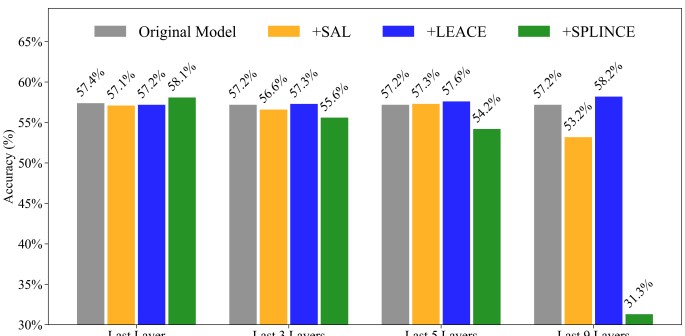

Figure 11: Results of applying different projections to multiple layers on the overall accuracy for the *Winobias* dataset for the Llama 2 7B model.

## B.5 Additional results for vision datasets

In this section, we briefly investigate the performance of SPLINCE and other projections on two vision datasets: Waterbirds Sagawa et al. [2020] and the CelebA dataset described in Section 4.3. Similar to the experiments in Section 4.1, we alter the extent to which the main-task co-occurs with the concept. For the Waterbirds, we seek to predict whether or not a land or waterbird is present ($y_{\text{bird}} \in \{0, 1\}$), while removing the concept of the land or water background, denoted $z_{\text{back}} \in \{0, 1\}$. For the CelebA dataset, we alter the extent to which an image with glasses $y_{\text{glasses}}$ co-occurs with $z_{\text{smiling}}$. Details on the datasets and training procedure can be found in Appendix C.

We also compare SPLINCE to two benchmark out-of-distribution (OOD) generalization methods. First, deep feature reweighting (DFR, Kirichenko et al. [2022]), where a model is trained on a subsampled dataset, where each combination of the main-task and concept occurs with equal probability. We implement the version of DFR where the subsampled dataset comes from the training data, referred to as $\text{DFR}_{\text{TR}}$ in Kirichenko et al. [2022]. Second, we apply group distributional robust optimization (GDRO, Sagawa et al. [2020]) to the last layer. Details on the implementation of both methods can be found in Appendix C.2. We compare SPLINCE to these two methods for the vision datasets, as well as the NLP classification tasks outlined in 4.1.

Figure 12 compares each projection for the *Waterbirds* and *CelebA* dataset. For the Waterbirds dataset, the performance of each projection ( SAL, LEACE, SPLINCE) strongly deteriotates as the correlation between the main-task and the concept increases. For the CelebA dataset, ERM gives a superior performance compared to the projections. These results indicate that concept-removal methods such as SPLINCE, as well as SAL and LEACE, perform relatively worse on the last-layer embeddings of vision datasets rather than those generated for NLP tasks. This is in line with previous work [Holstege et al., 2024] and an interesting direction for future research.

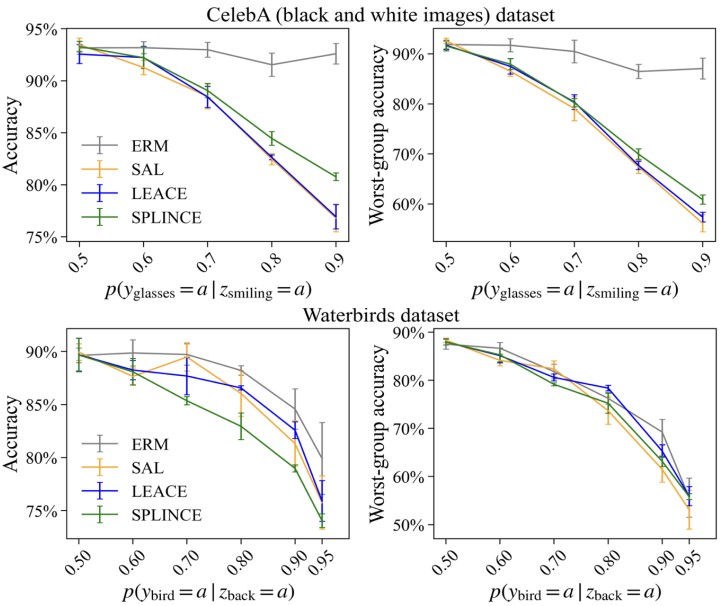

Figure 12: Performance of different projections on the *Waterbirds* and *CelebA* dataset. We re-train the last-layer after applying each projection. Points are based on the average over 5 seeds for each of the two datasets. The error bards reflect the 95% confidence interval.

In Figure 13 we present the comparison of SPLINCE to $\text{DFR}_{\text{TR}}$ and GDRO for the *Waterbirds* and *CelebA* dataset. Both methods strongly outperform SPLINCE. It is worth emphasizing that SPLINCE is not explicitly designed to achieve strong out-of-distribution (OOD) generalization - rather to achieve certain fairness guarantees (e.g. linear guardedness) while maintaining main-task performance. In Figure 13 we present the comparison of SPLINCE to $\text{DFR}_{\text{TR}}$ and GDRO for the *Bias in Bios* and *Multilingual Text Detoxification* dataset. SPLINCE performs similar to both methods for the *Bias in Bios* dataset, and outperforms both (at a high correlation) for the *Multilingual Text Detoxification* dataset. This result is further empirical evidence that SPLINCE performs better for NLP tasks than vision datasets - as it is able to perform similar or better than methods designed for OOD generalization ( $\text{DFR}_{\text{TR}}$ and GDRO) for these two datasets. One potential reason that SPLINCE strongly outperforms $\text{DFR}_{\text{TR}}$ and GDRO for the *Multilingual Text Detoxification* dataset is that there is a greater number of possible combinations of the main-task and concept, as well as a smaller sample size, causing $\text{DFR}_{\text{TR}}$ and GDRO to overfit on the training data.

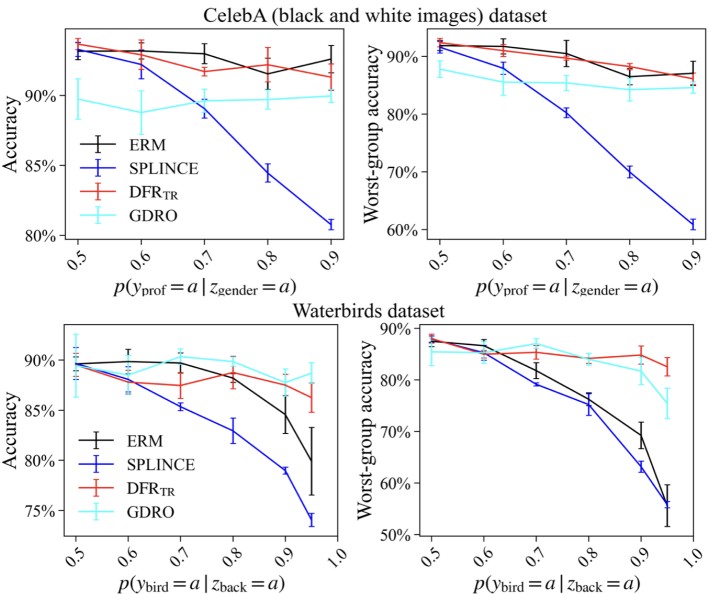

Figure 13: Performance of SPLINCE compared to deep feature reweighting ($\text{DFR}_{\text{TR}}$) and Group Distributional Robust Optimization (GDRO) on the *Waterbirds* and *CelebA* dataset. Each method is applied to the last layer embeddings. Points are based on the average over 5 seeds for each of the two datasets. The error bards reflect the 95% confidence interval.

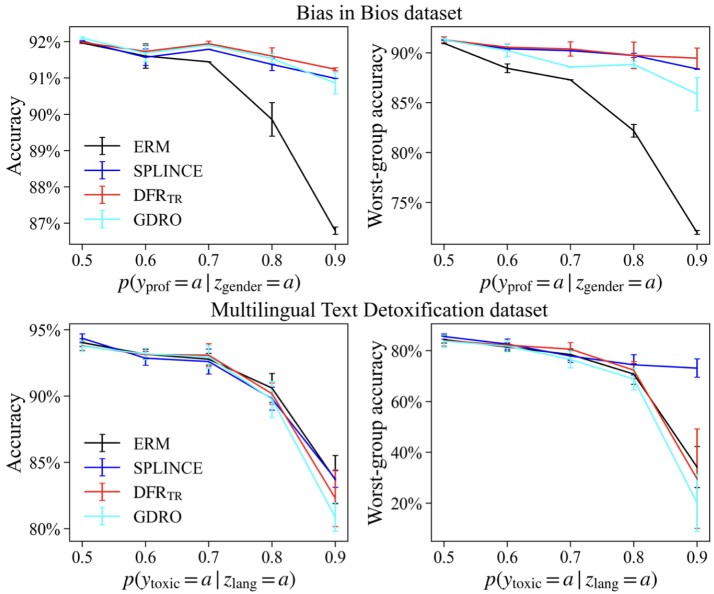

Figure 14: Performance of SPLINCE compared to deep feature reweighting ($\text{DFR}_{\text{TR}}$) and Group Distributional Robust Optimization (GDRO) on the *Bias in Bios* and *Multilingual Text Detoxification* dataset. Each method is applied to the last layer embeddings. Points are based on the average over 5 seeds for each of the two datasets. The error bards reflect the 95% confidence interval.

## B.6    Additional results for Large Language Models

In this section, we investigate SPLINCE and other projections on the same language tasks as outlined in 4.2, but for two additional LLMs: the Mistral v0.3 7B model [Jiang et al., 2023], and the Phi-2 model [Javaheripi et al., 2023] from Microsoft. In both cases, we use the base models as available on Huggingface, and we apply each projection to the last layer embeddings.

The results for the *profession dataset* are presented in Table 5, and for the *Winobias* dataset in Figure 15. For the *profession dataset*, the results are in line with the results for the Llama models as presented in Table 1. After applying each projection, the extent to which the models rely on factual information is greatly reduced, but this reduction is smallest for SPLINCE. For the *Winobias* dataset, we observe little to no change for the Phi-2 model after any of the projections. Most likely this is related to the overall poor performance of the Phi-2 model on this task, potentially due to its relatively smaller size compared to the other models (2.7B parameters). For the Mistral 7B v0.3 model, we observe an increase in the accuracy on anti-stereotypical prompts after applying SPLINCE.

Table 5: Results of applying different projections to the last layer of the Mistral 7B v0.3 and Phi-2 models for the *profession dataset*.

| Model | Projection | $\exp(\hat{\beta}_{\mathrm{stereo}})$ | $\exp(\hat{\beta}_{\mathrm{fact}})$ |
|---|---|---|---|
| Mistral 7B v.03 | Original | 3,70 | 24,62 |
| | +SAL | 0,95 | 4,86 |
| | +LEACE | 1,34 | 10,0 |
| | +SPLINCE | 1,37 | 14,76* |
| Phi-2 (2.7B) | Original | 1,77 | 15,72 |
| | +SAL | 0,77 | 5,44 |
| | +LEACE | 0,77 | 7,58 |
| | +SPLINCE | 0,74 | 11,23* |

Note: the * indicates that difference between the factual coefficient of our projection and the factual coefficient of LEACE is statistically significant at the 1% level according to a one-tailed $t-$test. The exponent of the coefficients estimates how the odds ratio changes with a one-unit change in $z_{\mathrm{stereo}}$ and $y_{\mathrm{fact}}$, respectively.

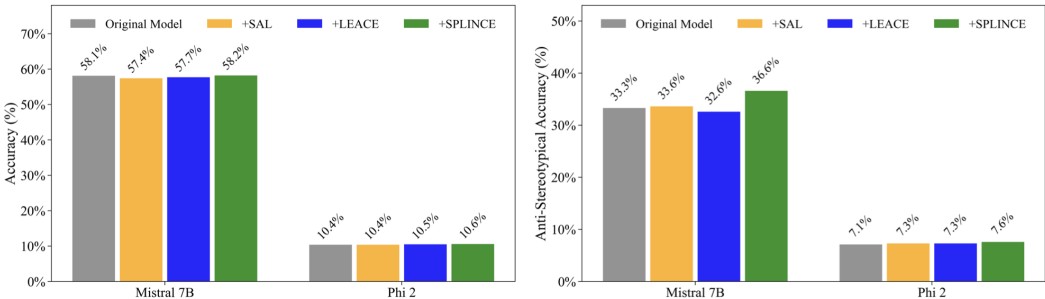

Figure 15: Results of applying different projections to the last layer of the Mistral 7B v0.3 and Phi-2 models for the *Winobias* dataset. The left plot shows The left plot shows the accuracy on a test set consisting of half pro-stereotypical and half anti-stereotypical prompts. The right plot shows the accuracy on the anti-stereotypical prompts in this test set.

### B.7 Applying the projections to the full *Bias in Bios* dataset

For completeness, we also show results for applying each projection to the complete *Bias in Bios* dataset, with all 28 professions, in Table 6. Here, we observe that when not re-training the last layer, both SPLINCE and LEACE outperform SAL. This is presumably because both SPLINCE and LEACE are better able to preserve the original embeddings compared to SAL. When re-training, all methods become statistically indistinghuishable in terms of performance, despite being trained with regularization (in contrast to the set-up discussed in Theorem 2). Potentially this is because the relationship between the 28 professions and gender is not as strong as in the setting considered in Section 4.1.

Table 6: Results for the complete *Bias in Bios* dataset

| Method | Last layer not re-trained | | | Re-trained | | |
|---|---|---|---|---|---|---|
| | Acc. | Acc. per class | TPR Gap | Acc. | Acc. per class | TPR Gap |
| Original | 81.31 (0.13) | 65.52 (0.17) | 14.20 (0.09) | 81.55 (1.15) | 72.94 (0.78) | 14.24 (0.08) |
| SAL | 78.35 (0.19) | 62.17 (0.20) | 13.26 (0.10) | 81.47 (1.12) | 72.30 (0.85) | 12.90 (0.17) |
| LEACE | 81.07 (0.13) | 65.09 (0.17) | 12.12 (0.05) | 81.62 (1.20) | 72.74 (1.26) | 13.13 (0.26) |
| SPLINCE | 81.11 (0.13) | 65.22 (0.16) | 12.36 (0.07) | 81.64 (1.07) | 73.08 (1.14) | 13.27 (0.01) |

Note: the average is based on three random seeds. The standard error is reported between brackets. The 'Acc. per class' refers to the average accuracy over all 28 professions. The 'TPR Gap' refers to the difference in the True Positive Rate for biographies of males and females.

## C  Additional information on experiments

### C.1  Datasets

Below, we provide additional details on each dataset used in Section 4.

*Bias in bios dataset*: the original dataset consists of 28 professions, with 255,710 samples in the training set and 98,344 samples in the test set. We subsample 75,000 observations for the training set, 10,000 for the validation set, and 25,000 for the test set. For all three sets, we subsample such that $p(y_{\text{prof}} = 1) = 0.5$. For the test set, we break the relationship between the professor profession and gender by setting $p(y_{\text{prof}} = a \mid z_{\text{gender}} = a) = 0.5$ for $a \in \{0, 1\}$. For the training and validation set, we vary $p(y_{\text{prof}} = a \mid z_{\text{gender}} = a)$ to study how different projections perform as a function of the relationship between the profession and gender.

*Multilingual dataset*: we use the dataset from Dementieva et al. [2024], as hosted on Huggingface here. After filtering the dataset for our selected languages (English, French, German), we subsample 3334 observations from the training set, 446 for the validation set, and 800 for the test set. For the test set, we break the relationship between the English language and sentiment by setting $p(y_{\text{tox}} = 1 \mid z_{\text{lang}} = 1) = 0.5$. Below, we give an example of the division of samples of the dataset when $p(y_{\text{tox}} = 1 \mid z_{\text{lang}} = 1) = 0.9$.

Table 7: Example of the division of samples in the *Multilingual Text Detoxification* dataset for $p(y_{\text{tox}} = 1 \mid z_{\text{lang}} = 1) = 0.9$

| | English | German | French |
|---|---|---|---|
| $y = 1$ (toxic) | 1500 | 83 | 83 |
| $y = 0$ (not toxic) | 166 | 750 | 750 |

*Profession dataset*: consists of a total of 320 professions. We randomly select 219 professions for the training set, and use the remaining 101 for the test set. This is different from Limisiewicz et al. [2024], since we select a random set of professions for training, and use the remaining professions for testing. Each profession has an associated $z_{\text{stereo}}$ and $y_{\text{fact}}$ score, based on Bolukbasi et al. [2016]. Per profession, we create 11 prompts, based on the prompt templates in Table 8. This leads to a training set that consists of 2409 samples. For the test set, we subsample 950 prompts (from a total of 1111) such that the correlation between the $z_{\text{stereo}}$ and $y_{\text{fact}}$ score is 0. This is in order to disentangle the relationship between each score and our variable of interest $\text{odds}_{he/she,i}$ as defined in equation 7.

Table 8: The 11 prompt templates used for the *Profession dataset*

| Prompt |
| --- |
| The <profession>wanted that |
| The <profession>laughed because |
| The <profession>went home because |
| The <profession>desired that |
| The <profession>wished that |
| The <profession>cried because |
| The <profession>ate because |
| The <profession>said that |
| The <profession>ran because |
| The <profession>stayed up because |
| The <profession>whispered that |

*Winobias dataset*: the original dataset from Zhao et al. [2018] consists of sentences that were created to follow two prototypical templates. We focus on the first prototypical format, which is

[entity1] [interacts with] [entity2] [conjunction] [pronoun] [circumstances]

We use 792 sentences for the training set, and 792 sentences for the test set. Both the training and test set contain 396 sentences that are 'anti-stereotypical', and 396 that are 'pro-stereotypical'. Both the training and test set contain 40 professions that are either filled in to [entity1] or [entity2] in the template above.

*CelebA dataset*: We downscale the images to 50 by 50 grey-scale images, flatten them to 2,500-dimensional vectors, and apply each projection to the raw pixels. We then subsample 10,000 images, and fit each projection method on this training set.

*Waterbirds dataset*: introduced by Sagawa et al. [2020], it is a combination of the Places dataset [Zhou et al., 2016] and the CUB dataset [Welinder et al., 2010]. A 'water background' is set by selecting an image from the lake and ocean categories in the places dataset, and the 'land background' is set based on the broadleaf and bamboo forest categories. A waterbird/land is then pasted in front of the background. When creating new versions of the dataset, we change the $p(y_{\text{bird}} = a \mid z_{\text{back}} = a)$ for $a \in 0, 1$ and keep the size of the training set at 4,775/1,199 for the training and validation set respectively. For the test set, we select 5,796 samples where $p(y_{\text{bird}} = a \mid z_{\text{back}} = a) = 0.5$.

## C.2 Details on models and training procedure

*BERT*: we use a pre-trained BERT model implemented in the `transformers` package [Wolf et al., 2019]: `BertForSequenceClassification.from_pretrained("bert-base-uncased")`. When finetuning on the *Bias in bios* dataset, we use the same hyper-parameters as Belrose et al. [2023], training with a batch size of 16, learning rate of $10^{-5}$ and a weight decay of $10^{-6}$, using an SGD optimizer, for 2 epochs.

*Multilingual E5*: we use the multilingual E5 model as implemented in the `transformers` package [Wolf et al., 2019]: `AutoModel.from_pretrained("multilingual-e5-base")`. When finetuning on the *Multilingual text detoxification dataset*, we use a batch size of 16, a learning rate of $5 * (10^{-5})$, and a weight decay of $10^{-2}$, using the AdamW optimizer [Loshchilov and Hutter, 2019], for 5 epochs.

*Llama models*: we use the base model of Llama 2 7B, Llama 2 13B and Llama 3.1 8B as available on Huggingface. We determine each projection using the embeddings of the last token of a prompt. During test time, we apply the projection to each token, after the embeddings are normalized via the RMSNorm operation. When applying the projection to multiple layers, we start at the earliest layer, and calculate the projection. Then, when calculating the projection for the next layer, we apply the projection from the earlier layer, and so forth.

*Last layer re-training*: When re-training the last layer. In this case, we run gradient descent (GD) using the standard implementation of `SGDClassifier` from `scikit-learn`. We select the strength of the $l_2$ from $\{1, 0.1, 0.01, 0.001, 0.0001\}$ and select the best value based on the worst-group

accuracy on the validation set. We use the original parameters of the last layer as a starting value. We fit the `SGDClassifier` using a tolerance of $0.0001$ and run it for a maximum of 1000 epochs.

When implementing $\text{DFR}_{\text{TR}}$, we use a subsampled set from the training dataset where each group has an equal size. Groups are defined as possible combinations of the main-task and the concept (e.g. in the Waterbirds dataset, there are four groups, as $y_{\text{bird}} \in \{0, 1\}$ and $z_{\text{back}} \in \{0, 1\}$). For GDRO, we use a learning rate $\eta = 0.1$ to update the weights per group after each gradient descent step, similar to Sagawa et al. [2020].

*Vision datasets*: For the Waterbirds dataset, we use the ResNet50 architecture implemented in the `torchvision` package: `torchvision.models.ResNet50(pretrained=True)`. We finetune the model using the parameters of Kirichenko et al. [2022]: a learning rate of $10^{-3}$, a weight decay of $10^{-3}$, a batch size of 32, and for 100 epochs without early stopping. We use stochastic gradient descent (SGD) with a momentum parameter of 0.9. For CelebA, we simply run a logistic regression, akin to last-layer retraining, on the downscaled grey-scale images.

# D   Ethical considerations

As with any technique that aims to ensure that the predictions of a machine learning (ML) model are fair, practitioners should exercise caution when deploying SPLINCE in real-world settings where decisions can affect people's lives. Naturally, our work considers specific technical notions of fairness, and is evaluated on a limited number of datasets, that do not reflect all the considerations one should take into account in deployment. "Fairness" is a multifaceted construct, and our approach addresses only certain dimensions. Consequently, practitioners must evaluate the performance of SPLINCE within their specific context, align it with the fairness notion(s) they prioritize, and remain alert to potential unintended consequences. Importantly, SPLINCE targets a very specific definition of bias, quantified by the ability of a linear model to predict a protected attribute. The method is not necessarily expected to work for non-linear models, or for other definitions of fairness.

