# OpenReview forum: "Preserving Task-Relevant Information Under Linear Concept Removal"
_NeurIPS.cc/2025/Conference — NeurIPS 2025 poster_

### Official Review · Reviewer_hbbQ · 2025-06-26

**Clarity:** 3
**Significance:** 2
**Originality:** 2
**Rating:** 4
**Confidence:** 3

**Summary:**

This paper introduces a post-hoc method that provably removes sensitive concepts from neural representations while exactly preserving their covariance with task-relevant labels, achieving effective concept removal with minimal distortion to task-relevant information.

**Questions:**

Q1: SPLICE demonstrates superior performance compared to baseline methods in two out of three tested Llama models in Fig. 3. Could the authors provide insight into the potential causes of this performance variation across different Llama model versions? To strengthen the generalizability of these findings, would the authors consider expanding the evaluation to include additional LLM architectures?

Q2: In the CelebA experiment, the difference maps for SPLICE show notable changes around the glasses regions. Intuitively, since the smile concept is being removed, changes should be concentrated primarily in the smile regions rather than the glasses regions. Could the authors clarify this observation?

**Ethical Concerns:**

["NO or VERY MINOR ethics concerns only"]

**Final Justification:**

The rebuttal clarifies that the paper fits within the scope of linear representation and will add experiments on other open-source LLMs. Therefore, I keep the current positive rating.

The paper will also include a dedicated ethics statement, so I update the ** Ethical Concerns** to *Minor*.

**Limitations:**

yes

**Quality:**

3

**Strengths And Weaknesses:**

### Strengths
-  The paper is well-written and easy to follow.
- It introduces SPLICE, a theoretically grounded oblique projection method that uniquely enforces both the removal of sensitive concepts and the preservation of task-relevant information.
- Extensive experimental results are provided across language and vision tasks.

### Weaknesses
- The paper addresses the linear concept erasure problem and assumes that task-relevant information can be represented through covariance.   However, these assumptions may not hold in complex representations within large models, where concepts are often encoded in a highly entangled manner. Experimental results further reveal that the proposed method performs less effectively when applied to intermediate layers.
- Some relevant works are missing in the comparison and discussion. In particular, methods that also aim to preserve task-relevant information during concept erasure, such as [1] and [2] ([2] is potentially considered concurrent work under the NeurIPS policy).

[1] Robust Concept Erasure via Kernelized Rate-Distortion Maximization, NeurIPS 2023

[2] Fundamental Limits of Perfect Concept Erasure, AISTATS 2025

---

> ### Author Rebuttal · Authors · 2025-07-31
>
> We thank reviewer hbbQ for their thorough examination of the paper and the suggested points of improvement. We also thank them for pointing out that the paper contains 'Extensive experimental results' and our approach is 'theoretically grounded'. Below, we address their feedback point-by-point.
>
> > The paper addresses the linear concept erasure problem and assumes that task-relevant information can be represented through covariance. However, these assumptions may not hold in complex representations within large models, where concepts are often encoded in a highly entangled manner. Experimental results further reveal that the proposed method performs less effectively when applied to intermediate layers.
>
> We thank the reviewer for highlighting this point and giving us the opportunity to clarify what we mean by "task‑relevant information". Our work fits within a broad line of research on linear *representation engineering*—that is, using linear algebra to manipulate neural network representations and control the information they encode. This area now includes dozens of studies, mostly on large language models but also on other neural architectures, and it has proved useful for applications ranging from bias mitigation to causal analysis. In particular, the "linear representation hypothesis" — the claim that many concepts are, to a large extent, linearly encoded — has become central to mechanistic interpretability [1, 2, 3]. Within this context, our contribution can be viewed as a natural extension of erasure methods: we show that it is possible not only to *remove* unwanted information, but also to *preserve* information that we wish to keep.
>
> At the same time, we acknowledge that many features are *also* non‑linearly encoded; hence, the linear representation hypothesis holds only approximately. As the reviewer points out, this limitation may explain the weaker performance we observe in middle layers. Nonetheless, we believe that providing a closed‑form solution for the linear case is valuable (the experimental section lists several concrete use cases). In future work we plan to extend our objective to the non‑linear setting via, for example, kernelization — an approach that has already proven effective in adapting other linear techniques [4].
>
> [1] Bolukbasi, T., Chang, K.-W., Zou, J. Y., Saligrama, V., and Kalai, A. T. Man is to computer programmer as woman is to homemaker? debiasing word embeddings.
> In Lee, D., Sugiyama, M., Luxburg, U., Guyon, I., and Garnett, R. (eds.), Advances in Neural Information Processing Systems, volume 29.
>
> [2] Vargas, F. and Cotterell, R. Exploring the linear subspace hypothesis in gender bias mitigation. In Proceedings of the 2020 Conference on Empirical Methods in Natural Language Processing (EMNLP), pp. 2902–2913, Online, November 2020.
>
> [3] Kiho Park, Yo Joong Choe, and Victor Veitch. 2024. The linear representation hypothesis and the geometry of large language models. In Proceedings of the 41st International Conference on Machine Learning (ICML'24), Vol. 235. JMLR.org, Article 1605, 39643–39666.
>
> [4]  Ravfogel, Shauli, et al. "Adversarial concept erasure in kernel space." Proceedings of the 2022 Conference on Empirical Methods in Natural Language Processing.
>
>
> > Some relevant works are missing in the comparison and discussion. In particular, methods that also aim to preserve task-relevant information during concept erasure, such as [1] and [2] ([2] is potentially considered concurrent work under the NeurIPS policy).
>
> We thank the reviewer for raising these relevant works. We will include both works in the related work section of an updated version of the manuscript.
>
> Our reasoning for not including methods such as [1] and [2] in our manuscript is that we focus on a) linear transformations that b) guarantee linear guardedness. We believe this focus is appropriate as methods that fulfill these criteria are similar in their fairness guarantees and computational resources. If we expand the comparison to methods beyond these criteria, we would have to include a whole range of concept-removal methods in addition to [1, 2] - see for instance several methods mentioned in the related work section - and significantly increase the scope of the paper.
>
> [1] Robust Concept Erasure via Kernelized Rate-Distortion Maximization, NeurIPS 2023
>
> [2] Fundamental Limits of Perfect Concept Erasure, AISTATS 2025
>
> > Q1: SPLICE demonstrates superior performance compared to baseline methods in two out of three tested Llama models in Fig. 3. Could the authors provide insight into the potential causes of this performance variation across different Llama model versions? To strengthen the generalizability of these findings, would the authors consider expanding the evaluation to include additional LLM architectures?
>
> It is true that there is no formal guarantee that SPLICE always significantly outperforms baseline methods, although the paper tries to make clear heuristically and empirically that it often does. The fact that SPLICE demonstrates superior performance compared to baseline methods in two out of three tested Llama models, should be seen in this light. We do not have a detailed explanation as to why there is variation in performance between the Llama 2 and 3.
>
> That being said, we agree with the suggestion of the reviewer to strengthen the generalizability of our findings. We are currently working on repeating the experiments in section 4.2 for other open-source LLMs, namely Mistral 7B and phi-2. We will include these experiments in an updated version of the manuscript.
>
>
> > Q2: In the CelebA experiment, the difference maps for SPLICE show notable changes around the glasses regions. Intuitively, since the smile concept is being removed, changes should be concentrated primarily in the smile regions rather than the glasses regions. Could the authors clarify this observation?
>
> We thank the reviewer for allowing us to clarify the results in section 4.3 of the paper. Figure 4 and Figure 8 (the latter is in Appendix B.3) show that SPLICE leads to a transformation of the image around both the eyes and the mouth. This is as expected. The change around the eyes is to preserve $\mathrm{Cov}(\boldsymbol{x}, y_{\mathrm{glasses}})$. The alterations around the mouth are in order to ensure linear guardedness with respect to the concept $z_{\mathrm{glasses}}.$

---

> > ### Comment · Reviewer_hbbQ · 2025-08-01
> >
> > Thank you for your thorough rebuttal addressing my concerns. I look forward to seeing the additional results included in the final manuscript.

---

### Official Review · Reviewer_ynPj · 2025-06-30

**Clarity:** 3
**Significance:** 3
**Originality:** 4
**Rating:** 5
**Confidence:** 3

**Summary:**

This paper introduces a method for concept erasure that aims to maximally preserve task-relevant information. For example, one might want a neural network to be insensitive to the concept of gender to mitigate gender bias. While existing approaches can remove the targeted concept, they often also degrade the performance on the main task by inadvertently discarding task-relevant information. The authors propose a method called SPLICE, which applies an affine transformation to the feature representations. This transformation ensures that the resulting features are *linearly guarded* against the target concept—i.e., they exhibit zero covariance with the concept to be removed—while preserving the covariance between the features and the main task. The paper includes a theoretical result showing that training a classifier on such transformed features, using any linearly guarded method and without regularization, yields the same classifier as training on the original features. The method's effectiveness is demonstrated in three settings: (1) bias removal from textual embeddings, where a linear layer is trained with regularization for classification; (2) language modeling, where the final layer is trained after projection; and (3) concept erasure from image data.

**Questions:**

- Please discuss which applications call for preserving main task performance (as in SPLICE) versus internal feature representations (as in LEACE).

- For each main assumption of SPLICE (e.g., $U^{\perp} \cap \text{colsp}(W\Sigma_{x,y}) = \{ 0 \}$), please provide theoretical or empirical justification for its realism.

- Please define “task-relevant information” and clarify how to choose a suitable proxy for it in practice.

**Ethical Concerns:**

["NO or VERY MINOR ethics concerns only"]

**Final Justification:**

This paper makes a valuable contribution to the concept erasure literature, both by deriving an interesting theoretical result regarding the family of linearly guarded methods and by presenting a new approach that demonstrates strong empirical performance. The authors’ rebuttal further reinforced my recommendation to accept.

**Limitations:**

The authors adequately addressed the limitations.

**Paper Formatting Concerns:**

No formatting concerns.

**Quality:**

3

**Strengths And Weaknesses:**

**Strengths**

* The paper is well-written and clearly presented, with helpful visualizations that aid in understanding the SPLICE method.
* It makes a valuable theoretical contribution by illustrating how the choice of concept erasure method, among those satisfying linear guardedness, should depend on the specific objective, such as preserving overall feature information or maintaining main task performance.
* The paper presents theoretical results characterizing when linear guardedness uniquely determines the resulting model, even if different projection methods are used.

**Weaknesses**
* The concept of "task-relevant information" is not clearly defined in the paper and appears to be conflated with the classification target variable.
* There is no discussion of potential bias in the task-relevant information, \$\mathbf{y}\$, which could itself be a result of biased data-generating processes. For instance, in the case of debiasing a classifier screening job applications (lines 27–32), the dataset may inherently reflect gender bias. Preserving covariance with such a task variable might inadvertently retain the very bias the method aims to remove, especially when applying SPLICE in earlier layers.
* The paper lacks a discussion of which applications might benefit more from preserving main task performance (as done by SPLICE) versus preserving internal representations (as in methods like LEACE).
* The assumption $U^{\perp} \cap \text{colsp}(W\Sigma_{x,y}) =\{0\}$ (line 133) is quite strong and may not hold in general.
* The theoretical justification and empirical performance of the method seem to apply primarily to the later or final layers of a model (line 323, Appendix B.4).

**Minor Weaknesses**

* The methods discussed in lines 73–74 may still satisfy linear guardedness, even if they do not explicitly mention it, unless a theoretical proof is provided to the contrary.
* Typo: “an projection” → “a projection” (line 167)

---

> ### Author Rebuttal · Authors · 2025-07-31
>
> We thank reviewer ynPj for their delicate consideration of the paper and feedback. We also thank them for stating that the paper is 'well-written and clearly presented', and 'makes a valuable theoretical contribution'. Below, we address their feedback point-by-point.
>
> > The concept of "task-relevant information" is not clearly defined in the paper and appears to be conflated with the classification target variable
>
> > Please define “task-relevant information” and clarify how to choose a suitable proxy for it in practice.
>
> We thank the reviewer for pointing this out and allowing us to clarify the notion of "task-relevant information". Ideally, we would want to preserve the features that are relevant for recovering the performance of the main task. In the LEACE paper, it is proven that *erasing* the covariance with the sensitive information variable prevents a linear model from recovering this information. Extending this link between covariance and information, here we want to make sure that the covariance with the classification target variable, $\mathrm{Cov}(\boldsymbol{x}, \boldsymbol{y})$, is *preserved*. We agree that alternative notions of ``task-relevant information'' are possible, but we believe that instantiation of this information by the covariance with the target variable is a natural extension of LEACE. Moreover, this notion admits a closed-form solution and is empirically useful, as we show in the paper. Note that under isotropic features, it is easy to prove that preserving $\mathrm{Cov}(\boldsymbol{x}, \boldsymbol{y})$ is sufficient to preserve performance for classifying $\boldsymbol{y}$. We will include a proof for this in an updated version of the appendix.
>
> > There is no discussion of potential bias in the task-relevant information, which could itself be a result of biased data-generating processes. For instance, in the case of debiasing a classifier screening job applications (lines 27–32), the dataset may inherently reflect gender bias. Preserving covariance with such a task variable might inadvertently retain the very bias the method aims to remove, especially when applying SPLICE in earlier layers.
>
> After the SPLICE (or LEACE, or SAL) projection the embeddings are linearly guarded for the spurious concept label $\boldsymbol{z}.$ This means that, restricted to linear models, the projected embeddings cannot be meaningfully used to predict the spurious label. This is also the case if the spurious label is predicted indirectly, via a linear model that predicts the main-task label $\boldsymbol{y}.$ The logic is that the projection also inevitably affects any linear prediction model for $\boldsymbol{y},$ in such a way that the correlation between the prediction of $\boldsymbol{y}$ and the spurious label $\boldsymbol{z}$ is broken.
>
> We emphasize that information about the spurious concept might still be present in the projected embeddings, but only in a non-linear fashion. This information can indeed be uncovered by a DNN if SPLICE is applied only to some earlier layers of the network, as the reviewer suggests. Therefore, if SPLICE is applied to earlier layers of the network, it should also be applied to all subsequent layers. This is done in the LEACE paper, where it is called scrubbing [1].
>
> In a revised version of the manuscript, we intend to communicate these points more clearly.
>
> > The paper lacks a discussion of which applications might benefit more from preserving main task performance (as done by SPLICE) versus preserving internal representations (as in methods like LEACE)
>
> > Please discuss which applications call for preserving main task performance (as in SPLICE) versus internal feature representations (as in LEACE).
>
> We thank the reviewer for allowing us to make the comparison between SPLICE and LEACE more explicit. The key difference between the two methods is that SPLICE fulfills the constraint of preserving $\mathrm{Cov}(\boldsymbol{x}, \boldsymbol{y})$, at the cost of minimizing $\mathbb{E}[||\boldsymbol{x} - \mathbf{P}\boldsymbol{x}||_2^2]$. This can be particularly useful if there is some sort of correlation between $\boldsymbol{y}$ and $\boldsymbol{z},$ which there typically is, leading LEACE to remove $\mathrm{Cov}(\boldsymbol{x}, \boldsymbol{y})$. One typically expects a bigger advantage of SPLICE compared to LEACE when this correlation is stronger. This is illustrated empirically in Appendix B.2. The additional constraint of SPLICE comes at a cost - it minimizes $\mathbb{E}[||\boldsymbol{x} - \mathbf{P}\boldsymbol{x}||_2^2]$ to a lesser extent. Our results in Appendix B.4 indicate that this can potentially pose a problem when applying SPLICE at middle layers.
>
> > The assumption  (line 133) is quite strong and may not hold in general
>
> > For each main assumption of SPLICE (e.g., ), please provide theoretical or empirical justification for its realism.
>
> We believe that the assumption $U^\perp \cap \mathrm{colsp}(W \Sigma_{\boldsymbol{x},\boldsymbol{y}}) = \mathbf{0}$ is not very strong.  It only says that the linear subspaces of the covariance directions $\mathrm{Cov}(\boldsymbol{x}, \boldsymbol{y})$ and $\mathrm{Cov}(\boldsymbol{x}, \boldsymbol{z})$ do not perfectly overlap. For the case of binary labels, this means that the covariance vectors are not the very same vector (up to a scaling factor). To be more concrete, for the Waterbirds dataset this would mean there are only landbirds on a land background and waterbirds on a water background. This is typically not the case in realistic datasets and would make any attempt to distinguish main-task and spurious features based on the labels futile. The assumption still allows for vectors with a high cosine similarity, i.e., we do not assume no overlap between the underlying features that encode the two concepts. We intend to clarify these subtleties around the assumption in the final version of the paper.
>
> > The theoretical justification and empirical performance of the method seem to apply primarily to the later or final layers of a model (line 323, Appendix B.4).
>
> We broadly agree with this limitation as raised by the reviewer. There is indeed a risk that SPLICE performs worse for middle layers, as it deprioritizes minimizing $\mathbb{E}[||\boldsymbol{x} - \mathbf{P}\boldsymbol{x}||_2^2]$ in favor of fulfilling an additional constraint. However, the results in Appendix B.4 also indicate that there are cases where SPLICE applied intermediate layers can be beneficial (such as in the Bias in Bios dataset, or the experiment with the Profession dataset).
>
> > The methods discussed in lines 73–74 may still satisfy linear guardedness, even if they do not explicitly mention it, unless a theoretical proof is provided to the contrary.
>
> We thank the reviewer for pointing this out. We will adapt the writing in our related work section, as well as our writing in line 214-217 to reflect this point.
>
> > Typo: “an projection” → “a projection” (line 167)
>
> We thank the reviewer for raising this typo and will adapt it in an updated version of the manuscript.
>
> [1] Belrose, N., Schneider-Joseph, D., Ravfogel, S., Cotterell, R., Raff, E., and Biderman, S. Leace: Perfect linear concept erasure in closed form. In
> Oh, A., Naumann, T., Globerson, A., Saenko,
> K., Hardt, M., and Levine, S. (eds.), Advances
> in Neural Information Processing Systems, volume 36, pp. 66044–66063.

---

> > ### Comment · Reviewer_ynPj · 2025-08-01
> >
> > Thank you for the rebuttal. My concerns were almost fully addressed, and previous misunderstandings were clarified. I look forward to the authors incorporating the definition and discussion of task-relevant information, as well as the SPLICE "scrubbing" technique applied across multiple consecutive layers, into the revised manuscript.
> >
> > One concern that I believe remains insufficiently addressed is Question 1:
> > > Please discuss which applications call for preserving main task performance (as in SPLICE) versus internal feature representations (as in LEACE).
> >
> > In the rebuttal, the authors primarily outline a theoretical advantage of SPLICE over LEACE in settings with strong correlation between $\boldsymbol{y}$ and $\boldsymbol{z}$. While this may be obvious, I believe it is still worth noting that, unlike LEACE, SPLICE is not readily applicable in settings where the main task is complex or not explicitly defined, such as language generation or for vision foundation models like CLIP and DINO. Moreover, SPLICE may inadvertently amplify reliance on other, unknown biases or spurious correlations unrelated to the concept being removed, potentially more so than LEACE. For example, in a job screening classifier, SPLICE could increase covariance between other harmful or discriminatory features and the main task, whereas LEACE would more conservatively preserve the overall representations. At the same time, the broader question of "whether preserving main-task covariance is optimal under some setting" is duly acknowledged in the limitations.
> >
> > Overall, I view this work as a valuable contribution to the concept erasure literature, and I maintain my “Accept” recommendation.

---

> > > ### Author Response · Authors · 2025-08-03
> > > **Response**
> > >
> > > Thank you for your followup and the positive assessment of our work! We agree that there may be settings where preserving the overall distance to the original representations is better, particularly when the practitioner is not aware of which concepts can influence the main task. We empirically showed that SPLICE improves at least some notions of bias -- e.g., in the language model experiment in section 4.2. But, we agree that in the presence of unobserved confounders, maintaining the covariance with the main task can actually be disadvantages. If one aims to optimize a specific fairness metric, for example, it is recommended to empirically measure the effect of the method on that metric. Following your suggestion, we will add a discussion on the effect of unobserved confounds to the final version of this work.

---

### Official Review · Reviewer_5q88 · 2025-07-01

**Clarity:** 2
**Significance:** 2
**Originality:** 2
**Rating:** 4
**Confidence:** 4

**Summary:**

This paper introduces SPLICE, a linear projection method designed to remove sensitive concepts from neural network embeddings while preserving task-relevant information. SPLICE extends previous methods LEACE by adding an additional constraint to preserve covariance between features and task labels. The authors validate their approach using classification datasets such as Bias in Bios and Multilingual Text Detoxification, as well as language modeling benchmarks, demonstrating improved retention of task-relevant information.

**Questions:**

- How exactly are the embeddings computed for language modeling tasks without retraining LLM parameters?

- Can you clarify the closed-form solutions and practical computation of SPLICE in language model experiments?

- Why were existing robust methods such as DFR, AFR, and SELF not compared in your experiments?

- What does the comma in Table 1 and following tables mean?

**Ethical Concerns:**

["NO or VERY MINOR ethics concerns only"]

**Final Justification:**

SPLICE introduce only a minor modification to LEACE based on the original LEACE paper, which limits the theoretical contribution. Empirically, the use of Task-Relevant Information is reasonable and potentially useful to real-world applications. Given this paper is a theoretical paper, I have updated my score accordingly while still maintain a borderline opinion.

**Limitations:**

- Incremental contribution with limited theoretical novelty.

- Insufficient clarity regarding implementation and computational details.

- Limited dataset size and lack of standard group robustness metrics for CelebA.

**Paper Formatting Concerns:**

No paper formatting concerns

**Quality:**

3

**Strengths And Weaknesses:**

Strengths:

- The idea of preserving task-relevant covariance while removing sensitive attributes is conceptually appealing.

- SPLICE provides theoretical justification for its methodology, including closed-form solutions.

- Experiments demonstrate potential benefits in preserving important information in debiasing tasks.

Weaknesses:

- The contribution is incremental, primarily extending LEACE by introducing an additional constraint $P \Sigma_{x,y} =  \Sigma_{x,y}$.

- The implementation details regarding last-layer retraining and language modeling are unclear, especially the calculation of features for language modeling.

- Experimental validation is limited, with no worst-group accuracy reported on the widely-used CelebA dataset.

- Lacks comparison with established group robustness methods like DFR, AFR, and SELF.

---

> ### Author Rebuttal · Authors · 2025-07-31
>
> We are grateful to reviewer 5q88 for the careful and considerate review of our paper, as well as for the suggestions for improvement, which we find extremely valuable. We also thank them for commenting that our approach is 'conceptually appealing'. Below, we address the feedback point-by-point.
>
> > The contribution is incremental, primarily extending LEACE by introducing an additional constraint
>
> We agree with the reviewer that the method we introduce extends LEACE by introducing an additional constraint. However, by doing so, we are (to the best of our knowledge) the first to consider preservation of main-task information in the context of linearly guarded concept removal. This is a well-known problem in the general field of post-hoc (linear) concept removal methods [see, e.g., references in line 26 of the manuscript], but our paper makes it clear that it is possible to address this problem in conjunction with the demand of linear guardedness.
>
> Furthermore, as a by-product our paper also advances the more general understanding of linear concept erasure methods. Theorem 2 of the manuscript shows that the range of the projection does not impact the predictions of linear models (which includes a logistic regression on last-layer embeddings) that are re-trained after the projection without regularization. Focusing on linearly guarded projections, this means that in this setting, methods like SAL, LEACE and SPLICE only improve on a basic orthogonal projection because of regularization. We consider these theoretical advances of our paper as valuable contributions to the existing literature about (linear) concept erasure.
>
> >   The implementation details regarding last-layer retraining and language modeling are unclear, especially the calculation of features for language modeling
>
> > Can you clarify the closed-form solutions and practical computation of SPLICE in language model experiments?
>
> We thank the reviewer for allowing us to clarify how the projections (SAL, LEACE, SPLICE) are applied to an LLM. We discuss the two cases from the paper.
>
> **When applied to the last layer**:  we apply the projections to the embeddings $\boldsymbol{x} \in \mathbb{R}^d$ before they are mapped to the logits of the LLM over the vocabulary. Let $\mathcal{V}$ be the vocabulary of the LLM, and $\mathbf{A} \in \mathbb{R}^{d \times |\mathcal{V}|}$  the weights of the last layer of the LLM. The logits at the last layer are now given by $h(\boldsymbol{x}) = \boldsymbol{x} \mathbf{P} \mathbf{A}$, where $\mathbf{P} \in \mathbb{R}^{d\times d}$ is the respective projection matrix (SAL, LEACE, or SPLICE).
>
> **When applied to the middle layers**: suppose we apply the projection to the last $k$ layers out of a total of $L$. Let $\boldsymbol{x}^{(L - k)} \in \mathbb{R}^{d_{L-k}}$ denote the output of  layer $L - k$. We then apply the projections to these embeddings as $ h(\boldsymbol{x}^{(L - k)}) = \boldsymbol{x}^{(L - k)}\mathbf{P}$. These transformed embeddings  $h(\boldsymbol{x}^{(L - k)})$ are then processed by layer $L - k + 1$, after which we again gather the embeddings and fit a projection. This is repeated until we get to the last layer.
>
> Note that the description above applies to the embeddings for a single token. As noted in the appendix, when determining the projection on our training data, we use the embeddings of the last token of a prompt. During test time, we apply the projection to each token. If the embeddings are normalized via the RMSNorm operation (as is the case for the Llama models considered), we apply the projection after the normalization.
>
> We will clarify our approach in an updated version of the manuscript.
>
> > How exactly are the embeddings computed for language modeling tasks without retraining LLM parameters?
>
> There is no need to re-train the LLM to calculate the embeddings. We can simply gather the embeddings during a forward pass for a prompt.
>
> > Experimental validation is limited, with no worst-group accuracy reported on the widely-used CelebA dataset.
>
> We thank the reviewer for raising this issue. In the original manuscript, we conducted the experiment on the black \& white images from the CelebA dataset in section 4.3 in order to visually illustrate the differences between the different projections. Per the suggestion of the reviewer, we have conducted an experiment where we alter the correlation between smiling and whether or not someone is wearing glasses in these images. See our reply to reviewer z2Ty for the results of this experiment, which we will add to the final version of the paper.
>
> > Lacks comparison with established group robustness methods like DFR, AFR, and SELF
> > Why were existing robust methods such as DFR, AFR, and SELF not compared in your experiments?
>
> We thank the reviewer for this suggestion. As stated in response to reviewer z2Ty, we emphasize that the goal of our suggested projection (or concept-removal methods in general) is not to have a state-of-the-art performance in terms of worst-group accuracy. It is to achieve fairness guarantees (via linear guardedness) and give  control over features used by DNNs.
>
> We have added a comparison with two popular group robustness methods. First, we use deep feature reweighting (DFR, see [1]), using the same training data available to SPLICE. This amounts to the $\mathrm{DFR}_{\mathrm{tr}}$ method introduced in [1]. Second, we apply group-distributional robust optimization (GDRO) to the last layer of the trained DNN.
>
> (Note that we only report here worst-group accuracy and not average accuracy, whose results are in line with worst-group accuracy, due to length constraints of the rebuttal. We will gladly provide the average accuracy tables during the discussion phase, if the reviewer wishes us to.)
>
> === Waterbirds dataset ===
> Worst-Group Accuracy (%)
> | Method  | 0.5 | 0.6 | 0.7 | 0.8 | 0.9 | 0.95 |
> |:-------|:-----:|:-----:|:-----:|:-----:|:-----:|:-----:|
> | ERM | 87.51 [86.46-88.56] | 86.66 [85.45-87.86] | 81.81 [80.28-83.34] | 76.26 [75.04-77.49] | 69.25 [66.66-71.84] | 55.6 [51.57-59.63] |
> | SPLICE | 87.95 [87.35-88.56] | 85.28 [83.84-86.71] | 79.16 [78.86-79.45] | 75.23 [73.14-77.32] | 63.18 [62.1-64.26] | 55.81 [55.2-56.42] |
> | $\mathrm{DFR}_{\mathrm{TR}}$ | 88.03 [87.16-88.9] | 85.0 [84.27-85.73] | 85.37 [84.05-86.68] | 84.19 [83.22-85.15] | 84.84 [83.09-86.59] | 82.56 [80.8-84.32] |
> | GDRO | 85.47 [82.78-88.16] | 85.3 [83.2-87.4] | 87.06 [86.1-88.03] | 84.02 [82.88-85.16] | 81.73 [79.09-84.37] | 75.45 [72.5-78.4] |
>
> === CelebA dataset ===
> Worst-Group Accuracy (%)
> | Method  | 0.5 | 0.6 | 0.7 | 0.8 | 0.9 |
> |:-------|:-----:|:-----:|:-----:|:-----:|:-----:|
> | ERM | 91.85 [90.96-92.74] | 91.71 [90.41-93.01] | 90.47 [88.2-92.74] | 86.47 [85.06-87.89] | 87.05 [84.96-89.15] |
> | SPLICE | 91.56 [90.57-92.56] | 87.93 [86.86-88.99] | 80.22 [79.39-81.05] | 69.96 [68.94-70.99] | 60.87 [59.95-61.79] |
> | $\mathrm{DFR}_{\mathrm{TR}}$ | 92.36 [91.65-93.08] | 90.98 [89.94-92.02] | 89.67 [89.31-90.04] | 88.22 [87.68-88.75] | 86.11 [85.12-87.1] |
> | GDRO | 87.78 [86.35-89.22] | 85.53 [83.2-87.85] | 85.38 [84.08-86.68] | 84.22 [82.25-86.19] | 84.58 [83.61-85.55] |
>
> === Bias in Bios dataset ===
> Worst-Group Accuracy (%)
> | Method  | 0.5 | 0.6 | 0.7 | 0.8 | 0.9 |
> |:-------|:-----:|:-----:|:-----:|:-----:|:-----:|
> | ERM | 90.96 [90.91-91.01] | 88.45 [88.01-88.89] | 87.29 [87.27-87.3] | 82.19 [81.54-82.83] | 71.99 [71.79-72.19] |
> | SPLICE | 91.28 [91.25-91.31] | 90.41 [90.39-90.43] | 90.22 [90.2-90.24] | 89.74 [89.54-89.95] | 88.39 [88.35-88.43] |
> | $\mathrm{DFR}_{\mathrm{TR}}$ | 91.31 [90.99-91.62] | 90.57 [90.26-90.87] | 90.39 [89.67-91.11] | 89.75 [88.42-91.08] | 89.47 [88.44-90.49] |
> | GDRO | 91.31 [91.18-91.44] | 90.23 [89.61-90.85] | 88.59 [88.54-88.63] | 88.84 [88.5-89.17] | 85.87 [84.21-87.54] |
>
> === Multilingual dataset ===
> Worst-Group Accuracy (%)
> | Method  | 0.5 | 0.6 | 0.7 | 0.8 | 0.9 |
> |:-------|:-----:|:-----:|:-----:|:-----:|:-----:|
> | ERM | 84.32 [82.94-85.69] | 81.45 [79.91-82.98] | 78.46 [76.18-80.74] | 70.77 [66.7-74.84] | 34.23 [26.13-42.33] |
> | SPLICE | 85.56 [84.5-86.63] | 82.57 [80.51-84.62] | 77.88 [75.43-80.34] | 74.42 [70.44-78.4] | 73.12 [69.5-76.75] |
> | $\mathrm{DFR}_{\mathrm{TR}}$ | 83.93 [81.76-86.1] | 82.17 [81.22-83.12] | 80.58 [78.02-83.13] | 72.31 [68.92-75.7] | 29.62 [10.04-49.2] |
> | GDRO | 83.74 [81.21-86.26] | 81.79 [79.53-84.05] | 76.54 [73.24-79.84] | 68.85 [64.61-73.09] | 20.19 [8.87-31.52] |
>
> The results show that SPLICE is strongly outperformed by DFR and GDRO for the Waterbirds and CelebA datasets. We emphasize that this is also true for the other concept-removal methods in our baseline, namely LEACE and SAL (see our reply to reviewer z2Ty for experimental verification). For the Bias in Bios dataset, SPLICE performs similar to both DFR and GDRO. For the Multilingual dataset, it outperforms both methods when there is a strong relationship between language and toxicity (0.9). As noted in our reply to reviewer z2Ty, we believe this discrepancy in performance between vision and language classification tasks is an interesting area of future research.
>
> [1] Kirichenko, P., Izmailov, P., and Wilson, A. G. Last layer re-training is sufficient for robustness to spurious correlations. In The Eleventh International Conference on Learning Representations, ICLR 2023.
>
> > What does the comma in Table 1 and following tables mean?
>
> The comma's are used to denote decimal points. We apologize for any confusion created by this convention.

---

> > ### Comment · Reviewer_5q88 · 2025-08-01
> >
> > Thank you for the detailed rebuttal. The additional experiments and explanation of some experimental setups indeed make the technical soundness much better. However, I am still concerned about the incremental extension on LEACE. I will keep the score unchanged.

---

> > > ### Author Response · Authors · 2025-08-03
> > > **Response**
> > >
> > > Thanks for your response. We agree that our approach builds on that foundation; however, our contribution tackles a distinct, previously unsolved problem: simultaneously removing linear information about a protected attribute Z while guaranteeing retention of predictive signal for label Y. Indeed, in the empirical setting, we show several use cases where SPLICE is *significantly* more effective than LEACE -- e.g., around 2X increase in the preservation of factual gender information in the language modeling experiment (4.2). We believe that the main  contributions of the paper are (ii) highlighting the need to preserve main-task information, and deriving a method to do so, and (ii) extensive empirical evaluation that studies when and to what extent this is effective, on diverse domains.
> > >
> > > Beyond the relation to LEACE, we hope that our rebuttal successfully addressed the rest of the issues you previously raised.

---

### Official Review · Reviewer_z2Ty · 2025-07-01

**Clarity:** 4
**Significance:** 2
**Originality:** 3
**Rating:** 4
**Confidence:** 4

**Summary:**

Linear concept removal aims to follow the linear representation hypothesis, i.e. concepts in a feature learner such as a neural network, exist as linear combinations of each other. By assuming this hypothesis to be true, techniques are primarily aimed at projecting out biased directions in the representation space.

The proposed method, Splice, follows related work in this research area, and as such, is a relevant method for concept debiasing. It aims to solve a constrained optimization problem derived from [1], but with an important distinction - task relevant information, i.e. features that correspond to the actual label at hand, must be preserved when learning the projection matrix (the operator P on the covariance of the features and label must be an identity transformation). Further, the authors show that the operator effect on the prediction is determined only by the choice of kernel, and independent of the range. Experiments demonstrate the usefulness of Splice on shortcut removal in language and image data.

[1] Belrose, Nora, et al. "Leace: Perfect linear concept erasure in closed form." Advances in Neural Information Processing Systems 36 (2023): 66044-66063.

**Questions:**

1. Minor comment first: This method is a namesake of a CLIP-based explainability method called Splice [1], which caused a great deal of confusion for me initially, and might cause future confusion as well. Please consider a different name.
2. The U and V matrices in Theorem 1 have orthonormal columns. I wonder if there are some connections to SVD and other matrix factorization techniques (U and V being the left and right singular vectors in this case). Your paper would then open up possibilities of spectral properties of such spurious features.
3. Since the authors adopt the term "concepts" explicitly, I can't help but mention a strong line of recent work in concept activation vectors that probe similar ideas, albeit not in the domain of shortcut learning. Please see [2]. Is there any particular reason the authors use the term "concepts"?
4. What also interests me is if the feature matrix represents some sort of alignment between modalities, e.g. CLIP features. How would P behave in this multi-modal setup? There is a risk that the covariance between the vision features and labels are preserved but not the text, and vice versa.

[1] Bhalla, Usha, et al. "Interpreting clip with sparse linear concept embeddings (splice)." Advances in Neural Information Processing Systems 37 (2024): 84298-84328.

[2] Fel, Thomas, et al. "A holistic approach to unifying automatic concept extraction and concept importance estimation." Advances in Neural Information Processing Systems 36 (2023): 54805-54818.

**Ethical Concerns:**

["NO or VERY MINOR ethics concerns only"]

**Final Justification:**

I have kept my final score unchanged and recommend acceptance, as I think the added constraint on the original LEACE loss is a novelty, and leads to some interesting consequences. I did not increase my score because of limited coverage and success of the proposed method on the group robustness paradigm. Second, I am not sure whether the authors should claim guarantees about "fairness", since none of the traditional fairness metrics have been evaluated on the paper. I have taken into account other reviewers' critique of the paper as well in my final decision.

**Limitations:**

Yes

**Quality:**

3

**Strengths And Weaknesses:**

Strengths:

1. This paper is extremely well written and easy to follow. Every result proceeds intuitively from a previous result, and all design choices are well motivated.
2. Both theorems are relevant results, and while Theorem 1 essentially builds entirely on [1], the added constraint is a significant difference in my view and counts as novel.
3. Intuitive experimental results on both language and image shortcuts. Figure 2 demonstrates Splice is more robust to WGA as opposed to other projectors. Figure 3 demonstrates that the harmful effect of the bias variable is mitigated better by Splice (although I would argue just based on the illustration provided that LEACE is also pretty satisfactory).

Weaknesses.

1. The most obvious weaknesses perhaps, are the reliance on binary labels to demonstrate usefulness (this is not entirely the paper's limitation, but rather the limitation of the entire field of shortcut learning), and the absence of multi-shortcut results, such as UrbanCars. See [2].
2. There are also no results on Waterbirds, which is the de-facto dataset for evaluating worst-group accuracy. Further, there seems to be no consistency in the experiments chosen. While WGA seems to be a metric to observe for language data, it is not re-used in the image data. Please consider having WGA for both modalities, and also adding Waterbirds.
3. An out-of-distribution evaluation set can be a set where the shortcut of interest is explicitly removed. Have the authors considered doing an experiment like this? For example, an OOD case of Waterbirds is when the eval set has landbirds exclusively with water backgrounds, and waterbirds exclusively with land backgrounds. What would be the interpretation of P in this case?
4. I am not convinced that this method works when a single shortcut of interest is unknown. At the very least, there should be a discussion on *where* exactly the spurious features lie. Based on how the optimisation is currently set up, I am curious whether the span of the learned P belongs to the span of the orthogonal complement of the spurious features. My hunch is it is, but a result explicitly investigating this connection would be extremely interesting. For example, see Theorem 2 in [3].

[2] Li, Zhiheng, et al. "A whac-a-mole dilemma: Shortcuts come in multiples where mitigating one amplifies others." Proceedings of the IEEE/CVF Conference on Computer Vision and Pattern Recognition. 2023.

[3] Ruiz Luyten, Max, and Mihaela van der Schaar. "A theoretical design of concept sets: improving the predictability of concept bottleneck models." Advances in Neural Information Processing Systems 37 (2024): 100160-100195.

---

> ### Author Rebuttal · Authors · 2025-07-31
>
> We are grateful to reviewer z2Ty for carefully examining the paper and their feedback.
> We also thank them for their remarks that the paper is 'extremely well-written', and that both theorems in the paper are 'relevant results'. Below, we address the feedback point-by-point.
>
> >The most obvious ... such as UrbanCars. See [1].
>
> We agree with reviewer z2Ty that this is a limitation of our approach. It requires access to both a concept label $z$ and a task of interest $y$. We also agree that this is a limitation of the current literature on concept-removal - virtually all existing approaches (see for instance [4, 5, 6, 7]) require a concept label $z$. Extending these approaches (including SPLICE) to a scenario where no (or limited) labels for $z$ are available is an interesting area of future research.
>
> While our paper does not include multi-shortcuts as suggested in [1], our suggested projection can be easily extended to this case. The $z$ will then include both the labels of the first spurious attribute (e.g., background) as well as the second spurious attribute (e.g., another co-occurring object).
>
> >There are also ... adding Waterbirds.
>
> We thank the reviewer for this suggestion. We emphasize that the goal of our proposed projection (or concept-removal methods in general) is not to have a state-of-the-art performance in terms of worst-group accuracy. Instead, it is to achieve certain fairness guarantees (via linear guardedness) and give the user control over features used by the deep neural network (DNN).
>
> Per suggestion of the reviewer, we have evaluated SPLICE, as well as the baselines in the paper (SAL, LEACE) on the Waterbirds dataset. We have also evaluated each projection in a similar manner on the black-white version of the CelebA dataset in Section 4.3. In the original version of the manuscript we did not consider this evaluation for this dataset, since it was only used to illustrate which features are removed by each projection.
>
> For Waterbirds, SPLICE performs similar to LEACE and SAL, although it is worth noting that all concept-removal methods perform poorly on this dataset (in comparison to standard group robustness methods - see our reply to reviewer 5q88). For CelebA, SPLICE outperforms LEACE and SAL.
>
> (Note that we only report here WGA and not average accuracy, whose results are in line with WGA, due to length constraints of the rebuttal. We will gladly provide the average accuracy tables during the discussion phase, if the reviewer wishes us to.)
>
> === Waterbirds Dataset ===
> Worst-Group Accuracy (%)
> | Method | 0.5 | 0.6 | 0.7 | 0.8 | 0.9 | 0.95 |
> |:-------|:-----:|:-----:|:-----:|:-----:|:-----:|:-----:|
> | ERM | 87.5 [86.5-88.6] | 86.7 [85.5-87.9] | 81.8 [80.3-83.3] | 76.3 [75.0-77.5] | 69.3 [66.7-71.8] | 55.6 [51.6-59.6] |
> | SPLICE | 88.0 [87.3-88.6] | 85.3 [83.8-86.7] | 79.2 [78.9-79.5] | 75.2 [73.1-77.3] | 63.2 [62.1-64.3] | 55.8 [55.2-56.4] |
> | SAL | 88.3 [87.9-88.8] | 84.1 [83.0-85.3] | 82.4 [80.7-84.0] | 73.7 [70.8-76.5] | 61.5 [58.8-64.1] | 53.1 [49.1-57.1] |
> | LEACE | 88.0 [87.4-88.6] | 85.1 [83.6-86.7] | 80.6 [79.9-81.3] | 78.4 [77.8-78.9] | 65.3 [64.1-66.5] | 55.9 [53.9-57.9] |
>
> === CelebA Dataset ===
> Worst-Group Accuracy (%)
> | Method | 0.5 | 0.6 | 0.7 | 0.8 | 0.9 |
> |:-------|:-----:|:-----:|:-----:|:-----:|:-----:|
> | ERM | 91.9 [91.0-92.7] | 91.7 [90.4-93.0] | 90.5 [88.2-92.7] | 86.5 [85.1-87.9] | 87.1 [85.0-89.2] |
> | SPLICE | 91.6 [90.6-92.6] | 87.9 [86.9-89.0] | 80.2 [79.4-81.0] | 70.0 [68.9-71.0] | 60.9 [60.0-61.8] |
> | SAL | 92.6 [92.0-93.1] | 86.5 [85.5-87.6] | 79.1 [76.6-81.5] | 67.4 [66.1-68.7] | 56.1 [54.4-57.9] |
> | LEACE | 91.7 [90.9-92.6] | 87.5 [86.0-89.0] | 80.4 [78.9-81.8] | 67.7 [66.9-68.5] | 57.4 [56.4-58.3] |
>
> Based on these results, it appears that SPLICE (as well as other concept-removal methods) performs much better for classification tasks in the language domain (Bias in Bios, Multilingual detoxification dataset) than in vision (Waterbirds, CelebA). We believe this limitation is an interesting direction for future research. We will include this limitation and the results for Waterbirds and CelebA in an updated version of the manuscript.
>
> > An out-of-distribution evaluation ... interpretation of $\mathbf{P}$ in this case?
>
> As said in our previous reply, the primary goal of SPLICE is not OOD generalization. Instead, its aim is linearly guarded concept removal while maintaining main-task performance. The experimental setup of OOD generalization is used to test to what extend SPLICE (and other existing methods) achieves this goal.
>
> That being said, the default setup of OOD generalization experiments is a training set with a spurious correlation and a test set without this spurious correlation. In the case of Waterbirds, this amounts to a test set with four equal-size subgroups. The evaluation metrics are typically the average accuracy and the worst-group accuracy. The reviewer suggests to work with a test set consisting of only two of the four subgroups: landbirds on water and waterbirds on land.
>
> It does not seem directly clear what this would add to the current experiment. The average accuracy now becomes less informative, as it neglects performance on the majority subgroups. Moreover, it is unlikely that this change will affect the worst-group accuracy, as the models typically perform the worst on one of the minority subgroups. Finally, the projection $\mathbf{P}$ is defined independently from the choice of the test set and its interpretation therefore is unaltered: it removes (linear) information about the spurious feature from the embeddings, while maintaining the covariance with the main-task label.
>
> > I am not convinced ... example, see Theorem 2 in [2].
>
> In reply to the first concern of the reviewer, it is true that SPLICE assumes knowledge of the labels associated with the spurious correlation (as acknowledged earlier in our reply). And this, of course, also assumes that the shortcut itself is known. We point out that this also holds for competing concept-removal methods like LEACE and SAL.
>
> Regarding the location of the spurious features, it is typically not true that the span/range of the learned $\mathbf{P}$ belongs to the span of the orthogonal complement of the spurious features. The kernel of $\mathbf{P}$ is spanned by the spurious features, but $\mathbf{P}$ is typically an oblique projection and not an orthogonal projection. This was already the case for LEACE, due to the whitening transformation which is shared with SPLICE. Following the framework of Thm. 2 in [2], this suggests a bias in the predictive model after projection of the last-layer embeddings. We agree with the reviewer that it would be very interesting to investigate this connection, possibly in the context of a bias-variance trade-off. We thank the reviewer for highlighting the connection with [2] and will include this in the final version of the paper.
>
> > Minor comment first: This ... a different name.
>
> We thank the reviewer for highlighting this namesake method and will consider to update the name from SPLICE to SPLINCE (Simultaneous Projection for LINear concept removal and Covariance prEservation) for the final version of the paper to avoid confusion.
>
> > The U and V matrices ... such spurious features.
>
> The fact that the matrices U and V in Theorem 1 have orthonormal columns is just for technical convenience. This choice of bases for U and V, respectively, is always possible and keeps Equation 4 simple. Furthermore, the matrices U and V act on the whitened features (see W in Eq. 4), whose covariance matrix has a trivial spectral decomposition. We therefore do not think that SVD (or other factorization techniques) applied to the matrices U and V would lead to new insights.
>
> > Since the authors adopt ... the term "concepts"?
>
> We adopt the term to be line with the literature on concept-removal methods. In this literature, a concept refers to a representation in the data of a human-defined object or phenomenon (in line with the original paper introducing concept activation vectors [8]). Concepts are usually represented via binary, one-hot encoded, or continuous variables, and often align with protected attributes (e.g., gender).
>
> > What also interests me ... and vice versa.
>
> We thank the reviewer for raising the question as to how our projection would function with CLIP. Suppose that for a text-image pair, we have the text features $\boldsymbol{t}$, and the image features $\boldsymbol{v}$. If for the text-image pair the concept labels $z$ and $y$ are available, one can simply apply our projection to both  $\boldsymbol{t}$ and $\boldsymbol{v}$. This approach appears to address the risk raised by the reviewer.
>
> The reviewer also raises an interesting point as to the interpretation of $\mathbf{P}$ if it is applied to a vector that represents alignment between modalities. In this case, SPLICE would work in a similar fashion, only now operating on the features where the text features $\boldsymbol{t}$ and vision features $\boldsymbol{v}$ align.
>
> [1] Li, Zhiheng, et al. "A whac-a-mole dilemma: Shortcuts come in multiples where mitigating one amplifies others." 2023
>
> [2] Ruiz Luyten, Max, and Mihaela van der Schaar. "A theoretical design of concept sets: improving the predictability of concept bottleneck models." 2024
>
> [3] Fel, Thomas, et al. "A holistic approach to unifying automatic concept extraction and concept importance estimation." 2023
>
> [4] Ravfogel, S., et al. Linear adversarial concept erasure. 2022
>
> [5] Belrose, N., et al. Perfect linear concept erasure in closed form. 2023
>
> [6] Ravfogel, et al. Null it out: Guarding protected attributes by iterative nullspace projection. 2020
>
> [7] Shun Shao, et al. Gold doesn’t always glitter: Spectral removal of linear and nonlinear guarded attribute information. 2023
>
> [8] Kim, B., et al. Interpretability beyond feature attribution: Quantitative testing with concept activation
> vectors (TCAV). 2018

---

> > ### Comment · Reviewer_z2Ty · 2025-08-01
> >
> > I would like to thank the authors for clarifying all my concerns. A few further comments:
> >
> > 1) My question on applying Splice to CLIP-aligned features was not satisfactorily addressed, as it is not simply using the text and vision features with their concept annotations. There exists a modal gap that confounds a lot of these representations. See [1]. However, this was not the main point of the paper, nor was this a main discussion point in the paper, and therefore, is not a "weakness" of the paper in my view. Just some things to think about. A very short paragraph explaining how the modal gap interacts for this method would be interesting.
> >
> > 2) The new results with Waterbirds and Celeb-A have opened up interesting new lines of enquiry for the authors. They observe that the language domain is more "friendly", which is in line with my previous comment. Images are hard! Please add these results to the main paper and a discussion to follow. In my view these new results are not a weakness of the paper, but makes it more interesting. But I do hope the authors recognise the nuance that "debiasing" does not necessarily imply robustness to spurious correlations.
> >
> > 3)  "Instead, it is to achieve certain fairness guarantees (via linear guardedness) and give the user control over features used by the deep neural network (DNN)" - I would be a bit more careful mentioning fairness guarantees, since this paper does not really test over direct fairness metrics such as disparate impact, demographic parity, etc.
> >
> > In general, my rating remains unchanged for now, but I have an overall positive assessment of this paper. I would be happy to update the rating further if the three points above are addressed well!
> >
> > [1] Liang, Victor Weixin, et al. "Mind the gap: Understanding the modality gap in multi-modal contrastive representation learning." Advances in Neural Information Processing Systems 35 (2022): 17612-17625.

---

> ### Author Response · Authors · 2025-08-02
> **Response**
>
> 1. Thank you for the follow-up! We want to be sure we have correctly understand your multimodal concern. Are you suggesting that a multimodal encoder might assign distinct latent directions to the same high-level concept across modalities -- for instance, CLIP allocating one direction to the visual expression of positive sentiment (a smile) and another to its textual expression (words such as happy)?
>  We agree that this cross-modal gap is intriguing. We remark that (i) This seems like a limitation of the underlying encoder. If the encoder itself does not align the modalities, our debiasing procedure can only operate on the representations it receives. (ii) The influence of such gap depends on the supervision. In our setting each training example carries a ground-truth label (e.g., “positive”), so the learned manipulation should follow whichever modality aligns with that label. When the modalities disagree, the method will naturally adjust the modality correlated with the labels and leave the other largely unchanged. We believe it is possible to test this hypothesis empirically -- for example, by selectively perturbing features in one modality-- and will discuss this issue in the final version.
>
> 2. We will add these results to the final version of this work. As our method is not causal in nature, we fully agree that debiasing without carefully considering confounds can even introduce new spurious correlations. We will improve the discussion of this point in the final version.
>
> 3. We fully agree with this observation and will discuss the disparity between representation-notions of debiasing and fairness outcomes. While we intuitively hope (and empirically test) that our interventions would improve certain notions of fairness, we acknowledge that different fairness notions exist (and are sometimes mutually exclusive), and that our objective does not directly translate into these fairness conditions.

---

> > ### Comment · Reviewer_z2Ty · 2025-08-04
> >
> > Hi, thanks for your response. I did not mean that CLIP would assign different directions to semantically related concepts. I meant that while representations are semantically aligned in CLIP, due to the existence of the modality gap, it is a hypothesis whether both the modalities are semantically aligned to your target label!
> > "We believe it is possible to test this hypothesis empirically -- for example, by selectively perturbing features in one modality-- and will discuss this issue in the final version." I agree. This would be an interesting result to analyze, and addresses my original question.
> >
> > In light of the multiple new results the authors have produced, and quite a few new points that need to be added to the paper in relation to the other reviews, I keep my original score unchanged.

---

### Author Response · Authors · 2025-08-09
**Concluding remarks**

As we are nearing the end of the discussion phase, we would like to reiterate that we are deeply thankful to all reviewers for their attentive consideration of the paper. We also would like to thank the reviewers for acknowledging the relevance and novelty of our work, as well as highlighting the idea of SPLICE is *conceptually appealing*, and that the paper *makes a valuable theoretical contribution*, is *extremely well written* and contains *intuitive experimental results on both language and image* data.

During the discussion phase, we have tried to address the concerns and questions of the reviewers. Based on the feedback of the reviewers, we will update the manuscript. Here is an overview of the key updates:
* we have added experimental results for the Waterbirds and (black-and-white) CelebA datasets, which are benchmark datasets to assess OOD generalization, comparing SPLICE with existing linearly guarded concept-removal methods LEACE and SAL [see reviewer z2Ty].
* we have added experimental results with the same setup, for both language and image data, comparing SPLICE with a baseline of concept-removal-methods with SOTA performance in terms of OOD generalization/group robustness: deep feature reweighting (DFR) and group-distributional robust optimization (GDRO) [see reviewer 5q88].
* in order to show that the performance of SPLICE is LLM-agnostic, we are currently performing similar experiments on other open-source LLMs (Mistral 7B, phi-2) [see reviewer hbbQ].
* we also added several clarifications to the text, including what we mean by 'task-relevant information' [see reviewer ynPj] and an underpinning of the assumptions of the SPLICE result of Theorem 1 [see reviewer ynPj].
* we have changed the acronym to SPLINCE (Simultaneous Projection for LINear concept removal and Covariance prEservation), in order to avoid confusion with a CLIP-based explainability method called Splice [see reviewer z2Ty].
* we have added a discussion about a possible vulnerability of SPLICE (compared to LEACE) when applied to multi-modal models, as due to the modality gap it is unclear whether both modalities are semantically aligned with the target label [see reviewer z2Ty].
* we have added an Impact Statement addressing possible ethical issues [see ethics reviewers WY37 and k2er].

Finally, we would also like to thank the reviewers for their engagement during the discussion period. We believe that as a consequence of their feedback and suggestions, the manuscript has greatly improved.

---

### Note · Authors · 2025-08-13

We would like to reiterate that we are deeply thankful to all reviewers for their attentive consideration of the paper. We also would like to thank the reviewers for acknowledging the relevance and novelty of our work, as well as highlighting the idea of SPLICE is conceptually appealing, and that the paper makes a valuable theoretical contribution, is extremely well written and contains intuitive experimental results on both language and image data.

During the discussion phase, we have tried to address the concerns and questions of the reviewers. Based on the feedback of the reviewers, we will update the manuscript. Here is an overview of the key updates:

we have added experimental results for the Waterbirds and (black-and-white) CelebA datasets, which are benchmark datasets to assess OOD generalization, comparing SPLICE with existing linearly guarded concept-removal methods LEACE and SAL [see reviewer z2Ty].
we have added experimental results with the same setup, for both language and image data, comparing SPLICE with a baseline of concept-removal-methods with SOTA performance in terms of OOD generalization/group robustness: deep feature reweighting (DFR) and group-distributional robust optimization (GDRO) [see reviewer 5q88].
in order to show that the performance of SPLICE is LLM-agnostic, we are currently performing similar experiments on other open-source LLMs (Mistral 7B, phi-2) [see reviewer hbbQ].
we also added several clarifications to the text, including what we mean by 'task-relevant information' [see reviewer ynPj] and an underpinning of the assumptions of the SPLICE result of Theorem 1 [see reviewer ynPj].
we have changed the acronym to SPLINCE (Simultaneous Projection for LINear concept removal and Covariance prEservation), in order to avoid confusion with a CLIP-based explainability method called Splice [see reviewer z2Ty].
we have added a discussion about a possible vulnerability of SPLICE (compared to LEACE) when applied to multi-modal models, as due to the modality gap it is unclear whether both modalities are semantically aligned with the target label [see reviewer z2Ty].
we have added an Impact Statement addressing possible ethical issues [see ethics reviewers WY37 and k2er].
Finally, we would also like to thank the reviewers for their engagement during the discussion period. We believe that as a consequence of their feedback and suggestions, the manuscript has greatly improved.

---

### Decision · Program_Chairs · 2025-09-17

**Decision:**

Accept (poster)

**Comment:**

This work focuses on the problem of removing unwanted concepts from model embeddings, while preserving task-relevant information and otherwise modifying the embedding to a minimal degree. Specifically, the aim is to make it so that the unwanted concept can not be linearly identified. The proposed approach,  SPLICE, addresses the issue that other post-hoc concept removal methods can inadvertently erase other information or concepts that are useful for the task at hand, which could otherwise lead to degraded task performance.  SPLICE addresses this by preserving covariance with the target task. This is achieved by constructing a projection that places the covariance between the representations and a protected attribute in its kernel, while maintaining the covariance between the representations and the -task label in its range. Further, the authors provide theoretical analysis that shows that training a classifier on the edited embeddings, using any linearly guarded method and without regularization, will result in learning the same classifier as training on the original embedding.

This paper is well written and clear, provides strong theoretical contributions, proposes a simple yet effective approach, and generally performs well on the empirical evaluations.

Several weaknesses were identified during the review process. Primarily, the requirement for concept and task labels,  a lack of comparison on Waterbirds (which has become a very common, "de facto" dataset for evaluating group accuracy) and worst group accuracy for CelebA, the approach being a straightforward extension of LEACE, some implementation details being unclear, lack of comparison to established group robustness methods ( DFR, AFR, and SELF), and concerns regarding the case where the task label and target concept are correlated.

The authors addressed most of these concerns during the rebuttal phase, particularly by adding new results on waterbirds and WGA for CelebaA, adding in a comparison to DFR, and clarifying the assumptions of their approach and behavior when task and concept are correlated.

After the rebuttal, some concerns remained. Novelty remained an issue for some reviewers, and while the new results on waterbirds are appreciated they show that the method may not method perform as well in the group robustness paradigm. However, regarding novelty, I agree with other reviewers that the primary strength and focus of this work is its new theoretical insights and argument that preserving main-task performance rather than minimizing change to the model's activations (as is done by other approaches) can often offer better utility. And regarding the WGA performance, I agree with reviewers that this is an interesting observation and should be noted by the authors in their limitation section, but does not negate the contributions of the work.

I agree with reviewers that this is valuable work that should be disseminated to the broader research community, and thus recommend acceptance. I urge the authors to follow through on their promise to the incorporate the recommendations of the ethics reviewers into a dedicated ethics statement in the final manuscript.